# A distinct Acyl-CoA binding protein (ACBP6) shapes tissue plasticity during nutrient adaptation in *Drosophila*

Xiaotong Li [1] & Jason Karpac [1] ✉

Nutrient availability is a major selective force in the evolution of metazoa, and thus plasticity in tissue function and morphology is shaped by adaptive responses to nutrient changes. Utilizing *Drosophila*, we reveal that distinct calibration of acyl-CoA metabolism, mediated by Acbp6 (Acyl-CoA binding-protein 6), is critical for nutrient-dependent tissue plasticity. *Drosophila* Acbp6, which arose by evolutionary duplication and binds acyl-CoA to tune acetyl-CoA metabolism, is required for intestinal resizing after nutrient deprivation through activating intestinal stem cell proliferation from quiescence. Disruption of acyl-CoA metabolism by Acbp6 attenuation drives aberrant 'switching' of metabolic networks in intestinal enterocytes during nutrient adaptation, impairing acetyl-CoA metabolism and acetylation amid intestinal resizing. We also identified STAT92e, whose function is influenced by acetyl-CoA levels, as a key regulator of acyl-CoA and nutrient-dependent changes in stem cell activation. These findings define a regulatory mechanism, shaped by acyl-CoA metabolism, that adjusts proliferative homeostasis to coordinately regulate tissue plasticity during nutrient adaptation.

Evolutionary diversity in body size across metazoan taxa, and sometimes even within species, is generated through changes in organ size and organ distribution relative to a complex body plan[1]. Consistently, organ size is determined by nutrition. Even after developmental growth, organs in adult metazoa dynamically remodel structure, size, and function in response to nutrient fluctuation[2,3]. This remodeling, often defined as tissue plasticity, is facilitated by changes in cell growth, cell polyploidization, cell differentiation, and cell proliferation (such as the ability of proliferative cells to shift between active, quiescent, senescent, and dormant states)[4–6]. Tissue plasticity is likely critical for acute and chronic maintenance of metazoan fitness.

Organ resizing and tissue remodeling can be influenced by an array of evolutionarily conserved and nutrient-dependent metabolic signaling pathways[7,8]. For example, signaling pathways dictated by insulin or target-of-rapamycin (TOR) can affect energy substrate availability and/or cell function to shape organ size across taxa[9,10]. However, nutrient fluctuations also drive adaptive changes in succinct, and often tissue-specific, metabolic networks—adjusting glucose vs. fat

vs. amino acid usage to redistribute cellular pools of metabolic substrates[11–14]. Beyond energy homeostasis, these substrates, like acetyl-coenzyme A (acetyl-CoA), nicotinamide adenine dinucleotide (NAD+), and S-adenosyl-methionine, are essential for cellular function[15–18]. This is due to their role in a wide-variety of molecular processes, from nucleotide/chromatin alterations to post-translational modification of proteins to facilitating enzymatic reactions[18]. Thus, it is critical to understand how unique metabolic pathways influence these distinct substrate pools to shape various cellular functions and control nutrient-dependent tissue plasticity.

Acetyl-CoA in particular is a crucial metabolic substrate for both histone and general protein (pan-)acetylation, able to subsequently adjust gene expression and cell signaling in response to nutrient cues[19–21]. The acetyl-CoA substrate pool used for acetylation is generally derived from carbohydrate sources through glycolysis, via mitochondria and the tricarboxylic acid (TCA) cycle[22,23]. However, nutrient fluctuations can drive the use of alternative sources for maintaining acetyl-CoA levels[24,25]. For example, the breakdown of

[1]Department of Cell Biology and Genetics, Texas A&M University, School of Medicine, Bryan, TX, USA. ✉e-mail: karpac@tamu.edu

stored lipids into fatty acids can provide acetyl-CoA for histones in order to modify chromatin during nutrient deprivation[26]. Moreover, nutrient availability will also influence the activity of enzymatic deacetylases and acetylases, concurrently with acetyl-CoA levels, which control cytoplasmic and nuclear protein acetylation patterns to dictate cellular function[27]. Expectedly, metabolic substrates such as acetyl-CoA often act as rheostats to fine-tune regulation of tightly controlled processes required for tissue plasticity, including cell proliferation and growth[28]. Specific metabolic pathways that adapt metabolic substrate pools to nutrient changes in order to shift cellular functions likely represent key 'sensors' and 'regulators' of nutrient-dependent tissue plasticity.

Acyl-coenzyme A (acyl-CoA) is one such substrate that is central to metabolic pathways which influence acetyl-CoA pools[29]. Acyl-CoA is generated by activation of fatty acids (fatty acyl-CoA), which are susceptible to oxidation and promote acetyl-CoA formation in mitochondria, driving oxidative phosphorylation (OXPHOS) and energy production[29,30]. Selective partitioning and movement of acyl-CoA relies on Acyl-CoA-binding proteins (Acbps)[31]. These proteins are evolutionarily conserved across eukaryotic kingdoms, and can bind acyl-CoA esters of various chain lengths[32]. Acbps have diverse functions, from acting as a neuropeptide (or diazepam-binding inhibitor [DBI]) to dictating food intake and systemic metabolism[32–37]. Furthermore, Acbps can also regulate cellular lipid metabolism and beta-oxidation of fatty acids, protecting acyl-CoA from hydrolysis to maintain partitioned acyl-CoA substrate pools[31]. To this end, mammalian ACBP binding to acyl-CoAs has been shown to facilitate fatty acid oxidation and cell proliferation (as opposed to senescence) in tumor formation based on nutrient availability and metabolic reprogramming within these tumorigenic cells[38,39]. Specific metabolic pathways such as those dictated by Acbp function and acyl-CoA metabolism have the potential to uniquely regulate cell fate decisions by adjusting both energy metabolism and metabolic substrate pools (like acetyl-CoA) to nutrient changes[40,41]. Exploring the various (and sometimes tissue-specific) regulatory nodes that integrate cellular metabolism during nutrient adaptation is likely key to understanding how unique metabolic pathways calibrate substrate pools to shape nutrient-dependent tissue remodeling.

Invertebrate genetic models provide unique advantages to explore the integration of cellular metabolism, nutrient availability, and tissue plasticity[5,40–42]. Here, we exploited multiple aspects of the fruit fly *Drosophila melanogaster* model and uncovered that acyl-CoA metabolism, distinctly mediated by Acbp6 (Acyl-coenzyme A binding-protein 6) rather than other Acbp paralogs, is critical for nutrient-dependent tissue remodeling. Leveraging phylogenetic and genetic analysis, we found that Acbp6 arose from a specific gene expansion event in *Drosophila* evolution. Acbp6 is specifically expressed in the *Drosophila melanogaster* intestine (and more specifically the midgut), a tissue with regenerative capacity that encompasses key nutrient-sensing and metabolic-regulatory functions[43]. We uncovered that Acbp6 is a key integrator of mitochondrial-dependent metabolic networks during essential nutrient adaptive events, specifically feeding-fasting-refeeding transitions. Nutrient-dependent changes in Acbp6 function and acyl-CoA metabolism influence acetyl-CoA metabolism and global cellular lysine acetylation in intestinal enterocytes, functionally differentiated epithelial cells that have nutrient-sensing and storage capabilities. The integration of these metabolic pathways in enterocytes is subsequently required for intestinal regrowth and remodeling (resizing) after nutrient deprivation and during refeeding through activating intestinal stem cell proliferation from quiescence. Dietary acetate supplementation can circumvent the need for Acbp6 in modulating acetyl-CoA metabolism, maintaining proliferative homeostasis to shape nutrient-dependent tissue remodeling. We also identified the STAT92e-Upd3 (cytokine) signaling axis, whose activity is influenced by acetyl-CoA levels and acetylation, as a key regulator of acyl-CoA and nutrient-dependent changes in stem cell quiescence and proliferation. These findings highlight that distinct calibration of acyl-CoA metabolism is crucial to adjust proliferative homeostasis and regulate tissue plasticity during nutrient adaptation.

## Results

### Acbp6, an integrative metabolic regulatory hub, was generated by gene expansion in the *Drosophila* genus with specialized expression in the midgut

In order to identify unique metabolic pathways or networks that could act as integrative regulatory nodes in tissue plasticity, we employed a small-scale meta-analysis of accessible *Drosophila melanogaster* transcriptomes. These transcriptomes highlight global gene expression changes in response to internal (such as aging) or external (such as infection) cues that are known to induce general tissue remodeling[44–48]. Exploiting these datasets, we found acyl-CoA binding-protein 6 (Acbp6) expression to be uniquely regulated in response to all of these stimuli (Supplementary Fig. 1a). Moreover, whole-genome expression analyses in *Drosophila* have also uncovered changes in *Acbp6* expression in response to starvation and high-sugar diet treatment[49], suggesting that Acbp6 is regulated by dietary and/or nutritional cues. Acbp6 is part of a family of proteins with evolutionarily conserved ACB (acyl-CoA binding) domains, allowing for acyl-CoA binding and transport[31]. After exploring genomic sequences across *insecta* orders, we found that species within the *Drosophila* genus have more *Acbp* genes compared to other insects (Fig. 1a and Supplementary Fig. 1b). Eight *Acbp* genes were identified in *Drosophila melanogaster*, and phylogenetic analysis revealed that *Acbp* genes underwent a remarkable extension during evolution (Fig. 1a). Moreover, *Acbp6* is tandemly arrayed with *Acbp2*, *Acbp3* and *Acbp4* within the *Drosophila melanogaster* genome. The expansion of the *Acbp* gene family in *Drosophila* also underlies specialized gene expression. Using available expression data resources for flies, we learned that Acbps expression is divergent, as *Acbp1*, *Acbp2*, *Anox*, and *CG8814* are widely-expressed across tissues, while *Acbp3*, *Acbp4*, *Acbp5*, and *Acbp6* are specifically expressed in the adult fly midgut and hindgut (Fig. 1b). The midgut, part of the larger gastrointestinal tract, is the major site of food digestion, nutrient absorption, and energy substrate storage, and to this end displays a high degree of plasticity in response to nutritional changes[5,50]. Combined with the phylogenetic evidence, these analyses suggest that Acbp6 may represent a key metabolic regulatory node in the evolution of nutrient adaptation.

### Acbp6 distinctly regulates midgut tissue plasticity during nutrition adaptation by adjusting proliferative homeostasis

In order to further explore Acbp6 expression/regulation in the *Drosophila* midgut during nutrient adaptation, we generated an in vivo expression reporter (using a 1,000 base pair promoter region upstream of *Acbp6* transcription start site, linked to red fluorescent protein [RFP]; Acbp6-RFP). Acbp6-RFP reporter activity is relatively low when flies are fed *ad libitum* (Fig. 1c). However, RFP intensity in the posterior midgut is increased in response to nutrient deprivation (2 days fasting), and subsequently normalized by refeeding (2 days, Fig. 1c, Supplementary Fig. 2a). Acbp6 reporter activity is mainly restricted to the large, polyploid cells of the midgut epithelium (functionally differentiated enterocytes; NP1Gal4, UAS-nlsGFP+ cells, ECs, Supplementary Fig. 2b), which project microvilli into the lumen and drive nutrient absorption. This nutrient-dependent pattern of Acpb6-RFP reporter activity is specific to the posterior midgut, the region where the intestinal stem cells are most active during regeneration, as reporter activity does not change in the anterior or middle midgut, the other regions that display Acbp6-RFP reporter activity (Supplementary Fig. 2a–d). To confirm reporter accuracy, we monitored *Acbp6* transcription in dissected posterior midguts (via qRT-PCR) during fasting-refeeding transitions, and again found that *Acbp6* transcript is induced during fasting and normalized upon refeeding

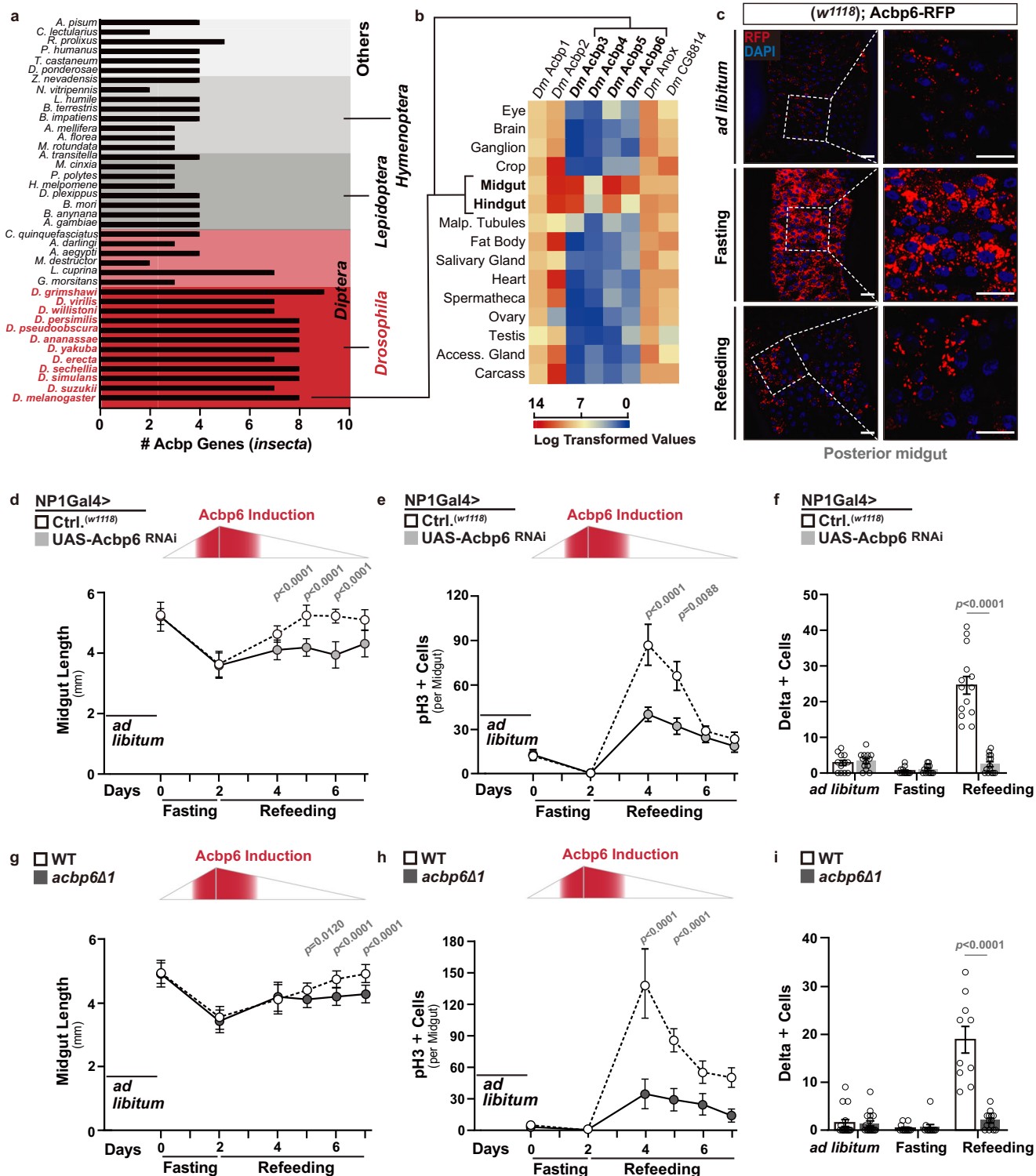

(Supplementary Fig. 2e). Finally, we generated an Acbp6$^P$-ACBP6$^{V5}$ transgenic fly expressing a C-terminal V5-tagged ACBP6 protein under the control of an Acbp6 promoter/enhancer region (Acbp6$^P$, similar to the RFP reporter). Immunostaining confirmed that ACBP6-V5 protein level is up-regulated during fasting (with localization in the cytoplasm of posterior midgut enterocytes) and normalized upon refeeding (Supplementary Fig. 2f, g). Altogether, these data provide evidence that Acbp6 is dynamically regulated during nutrient adaptation.

To determine if Acbp6 and potentially acyl-CoA metabolism play a role in nutrient-dependent tissue plasticity, we monitored midgut size during these nutrient adaptive responses (fasting-refeeding

transitions). The adult *Drosophila* midgut is extremely malleable, and can shrink and regrow/remodel (tissue resizing) in response to external cues, including nutrient fluctuations or dietary changes[2,7]. In control flies, we found that the midgut dramatically shrinks in response to nutrient deprivation (reducing length by approximately 30% after 2 days fasting), but resizes, back to normal length, 3–4 days after refeeding (Fig. 1d and Supplementary Fig. 3a, b). However, attenuating Acbp6 in enterocytes utilizing RNAi (NP1Gal4>UAS-Acbp6$^{RNAi}$) limits midgut resizing only during refeeding (Fig. 1d and Supplementary Fig. 3a–c), suggesting Acbp6 plays a key role in midgut plasticity during nutrient adaptation.

**Fig. 1 | Acbp6 regulates midgut tissue plasticity during nutrition adaptation by adjusting proliferative homeostasis. a** Histogram depicting the number of *Acbp* genes per genome of various insects. Left y-axis label displays species, right y-axis label displays orders. **b** Expression pattern of *Drosophila melanogaster Acbp* genes in different adult tissues; log transformed values from modEncode datasets. **c** Acbp6 expression pattern in dissected posterior midguts from *w1118*; Acbp6-RFP/ Acbp6-RFP transgenic flies during nutrient adaptation; stained with DAPI (blue); RFP (red) (*n* = 3 independent experiments). **d**–**f** Quantification of **d** midgut length (from left to right, bars represent *n* = 14, 15, 16, 14, 15, 18, 19, 17, 13, 20, 15, and 20 independent samples), **e** phospho-Histone (H3) positive cells (per whole dissected midgut, from left to right, bars represent *n* = 13, 13, 14, 14, 10, 19, 14, 10, 23, 14, 17, and 18 independent samples), and **f** Delta positive cells (per field, posterior midgut) upon enterocyte-specific depletion of Acbp6 during nutrient adaptation (from left to right, bars represent *n* = 12, 13, 12, 13, 14, and 14 independent samples).

Genotypes; *w1118*; NP1Gal4/+ (controls, Ctrl.), *w1118*; NP1Gal4/UAS-Acbp6 RNAi. **g**–**i** Quantification of **g** midgut length (from left to right, bars represent *n* = 30, 28, 30, 34, 29, 35, 17, 23, 22, 15, 18, and 16 independent samples), **h** phospho-Histone (H3) positive cells (per whole dissected midgut, from left to right, bars represent *n* = 11, 11, 11, 11, 11, 11, 19, 15, 17, 18, 13, and 11 independent samples), and **i** Delta positive cells (per field, posterior midgut) in *acbp6* mutant flies (*acbp6Δ1*) or controls (wild-type, WT; revertant) during nutrient adaptation (from left to right, bars represent *n* = 16, 17, 12, 11, 10, and 14 independent samples). Bars represent mean ± SEM (unpaired 2-tailed Student's *t*-test) **d**–**i**. The exact *p* values are provided in figure. Source data are provided as a Source Data file. Fasting represents 2 days of nutrient deprivation, and Refeeding represents 2 days of refeeding unless otherwise indicated. Diagram in panels **d**, **e**, **g**, and **h** represent an Acbp6 reporter induction strength timeline. Scale bars, 10 μm.

The midgut epithelial layer mainly contains four types of cells: intestinal stem cells (ISCs), enterocytes, secretory enteroendocrine cells (EEs), and enteroblasts (EBs, a postmitotic cell that can differentiate as an EC)[50]. ISC activation promotes midgut regeneration of functional enterocytes (through ISC proliferation/asymmetric division and differentiation of EBs) after tissue damage or significant EC loss, and ECs can non-autonomously dictate the proliferative status of ISCs through a variety of cues[51–54]. ISCs can be identified by the expression of Delta (Dl), a ligand for the Notch receptor, which activates the Notch signaling pathway in neighboring EBs to promote differentiation[51,52,55]. Midgut plasticity is thus shaped by epithelial remodeling, balancing cell gain and cell loss through control of ISC proliferative homeostasis. To this end, we uncovered that in control flies midgut ISCs adjust from inactive (quiescent) to active during the transition from fasting to refeeding, realized by increases in ISC proliferation (phospho-HistoneH3 [pH3]+ cells), and activation of ISC division and expansion (Dl+ cells) during the first 2-3 days of refeeding, before ISCs eventually start to return to quiescence (Fig. 1e, f and Supplementary Fig. 3d). Resizing of the midgut in response to nutrient adaptation is thus coupled to proliferative homeostasis. Similar to the changes in midgut length, Acbp6 function in enterocytes is required for ISC activation during refeeding, as attenuating Acbp6 (NP1Gal4>UAS-Acbp6RNAi) blocks increases in ISC proliferation and number (Fig. 1e, f and Supplementary Fig. 3d). Comparable results were obtained utilizing a temperature sensitive driver to attenuate Acbp6 only in the adult enterocytes (NP1Gal4,TubGal80ts > UAS-Acbp6RNAi, Supplementary Fig. 3e–f), and RNAi-mediated Acbp6 knockdown does not influence food intake rates during fasting-refeeding transitions (Supplementary Fig. 3g).

Furthermore, we generated an Acbp6 mutant fly line using CRISPR-Cas9 genome engineering in order to precisely target Acbp6 (leading to a distinct frameshift mutation in Acbp6; *acbp6Δ1*, Supplementary Fig. 4a–c). This mutant is viable, producing fertile homozygote adults (Supplementary Fig. 4d, e). *acbp6Δ1* homozygote mutant flies also display a deficiency in midgut plasticity in response to nutrient adaptation, highlighted by attenuation of both midgut resizing and ISC activation uniquely during fasting-refeeding transitions compared to revertant wild-type controls (Fig. 1g–i). *acbp6Δ1* heterozygote mutant flies display similar (although attenuated) phenotypes compared to homozygote mutants (gene-dosage effect), and re-expressing Acbp6 (UAS-Acbp6) can rescue the loss of ISC activation in *acbp6Δ1* heterozygote mutants (Supplementary Fig. 4f–i). This highlights that these plasticity phenotypes are, indeed, influenced by Acbp6 function.

These loss-of-function phenotypes also underlie specificity in Acbp6 function during refeeding and resizing, as Acbp6 does not influence adult midgut sizing from development (ad libitum) or shrinkage during fasting (Fig. 1d, g). Acbp6 also does not influence shifts in proliferative homeostasis from steady-state (*ad libitum*) to nutrient deprivation, as ISCs appear to revert to a further state of inactivity in response to fasting (Fig. 1e, f, h and i).

Related to other Acbp paralogs specifically expressed in the *Drosophila* intestine, attenuation of Acbp3 or Acbp5 specifically in enterocytes (NP1Gal4>UAS-Acbp3RNAi or UAS-Acbp5RNAi) did not impede ISC proliferation during refeeding, whereas Acbp4 function in enterocytes is essential for pupal-adult developmental transitions (eclosion; NP1Gal4>UAS-Acbp4RNAi, Supplementary Fig. 5a–c). These data indicate that enterocyte-derived Acbp6 likely plays a distinct role in regulating midgut remodeling during nutrient adaptation. To this end, we explored potential nutrient-dependent regulators of Acbp6 expression. By performing a comparative analysis of the promoter/ enhancer regions of the *Acbp3*, *Acbp4*, *Acbp5*, and *Acbp6* genes, we identified a DNA binding motif (characterized as CTACCAA) for the transcriptional regulator Schlank located uniquely in the upstream promotor region of *Acbp6* (Supplementary Fig. 5d). Schlank is a conserved ceramide synthase, which functions as both an enzyme and a nutrient- and homeodomain-dependent transcriptional regulator, often acting as a sensor for lipid levels to subsequently modulate gene expression[56]. Reducing Schlank function in enterocytes strongly decreases Acbp6-RFP reporter activity in the posterior midgut during fasting conditions (Acbp6-RFP, NP1Gal4>UAS-SchlankRNAi; Supplementary Fig. 5d, e). We also found that attenuation of Schlank (NP1Gal4>UAS-SchlankRNAi) impedes ISC proliferation during fasting-refeeding transitions (Supplementary Fig. 5f). Finally, investigating the transcriptional regulatory role of Schlank on different *Acbp* paralogs revealed that only *Acbp6* expression is diminished following Schlank attenuation in enterocytes (Supplementary Fig. 5g). Collectively, these findings indicate that the ceramide synthase Schlank serves as an upstream regulator of Acbp6 during nutrient adaptation in the midgut.

Taken together, our results reveal that Acbp6 function in enterocytes is essential for midgut tissue plasticity during nutrient adaptation. Specifically, Acbp6 is required for intestinal resizing after nutrient deprivation (fasting-refeeding transitions) through activating intestinal stem cell proliferation from quiescence and thus adjusting proliferative homeostasis. These data also show that Acbp6 induction (during fasting through nutrient-responsive transcriptional regulators) precedes its functional requirement for tissue resizing after refeeding (Fig. 1d–i), suggesting that distinct spatio-temporal calibration by Acbp6 is critical for nutrient-dependent midgut plasticity.

## Nutrient-dependent metabolic networks in the midgut are shaped by Acbp6 function

We next wanted to explore how Acbp6 and acyl-CoA metabolism influences nutrient-dependent metabolic networks in midgut enterocytes that could subsequently drive changes in tissue remodeling and proliferative homeostasis. Acbps are involved in selective partitioning of acyl-CoA, and can thus dictate acyl-CoA utilization for beta-oxidation in mitochondria or de novo lipid synthesis metabolic pathways in the cytoplasm[31] (Fig. 2a). In control flies, we found that changes in midgut neutral lipid storage (indicative of lipid synthesis) are minimal during fasting-refeeding transitions (Fig. 2b, Supplementary

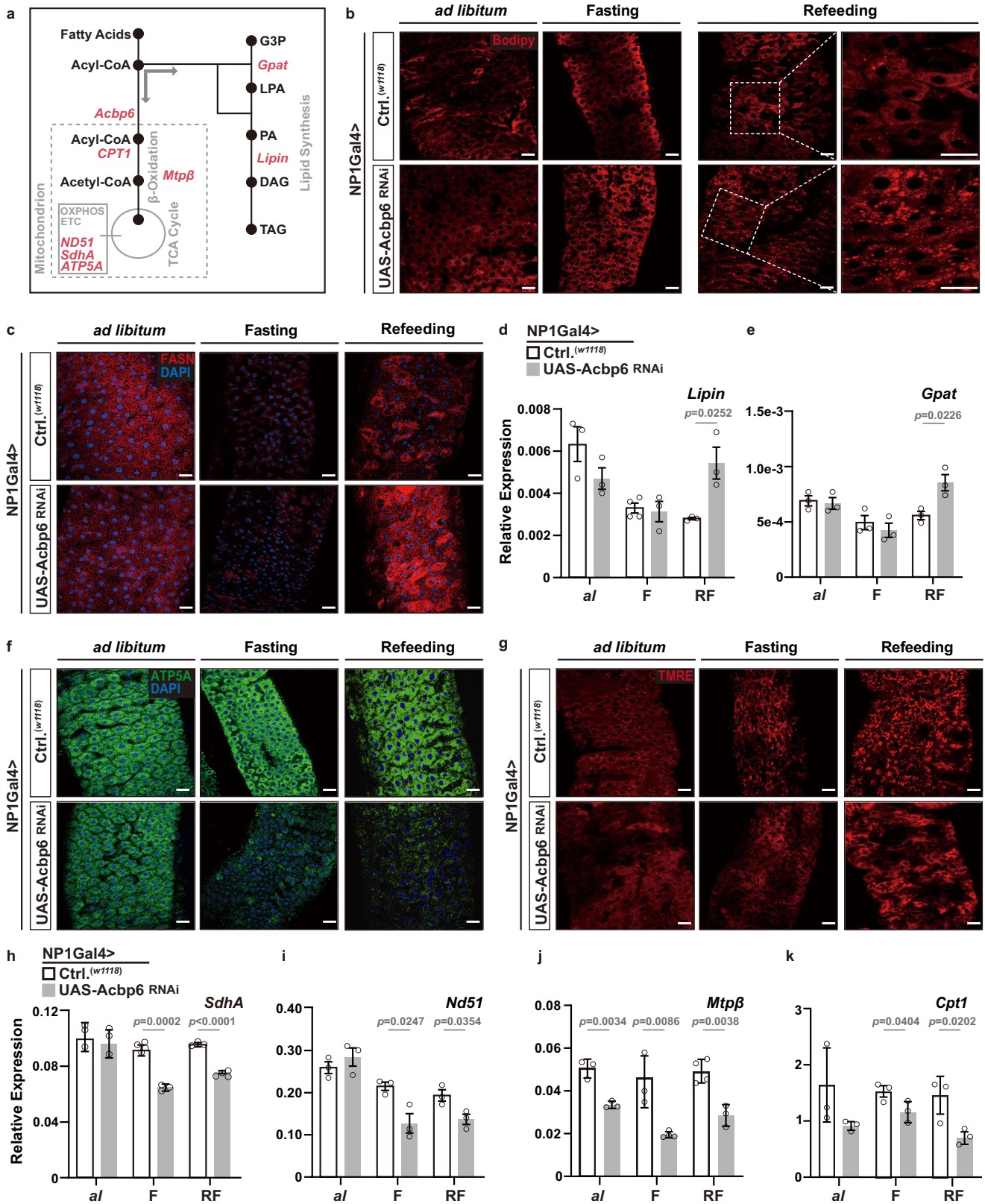

Fig. 6a). This is also true of Fatty acid synthase 1 (FASN1) levels, a rate-limiting enzyme in lipid synthesis, which decrease during fasting and are restored to normal during refeeding (Fig. 2c, Supplementary Fig. 6b-c). However, attenuating Acbp6 in enterocytes (NP1Gal4>UAS-Acbp6^RNAi) leads to increases in neutral lipid storage and elevated FASN1 levels in large, polyploid enterocytes, most prominently changed after refeeding (2 days, Fig. 2b, c, Supplementary Fig. 6a, b).

Additionally, Acbp6 attenuation also leads to transcriptional upregulation, in the midgut and uniquely during refeeding, of genes essential for lipid synthesis from acyl-CoA pools (Fig. 2d-e, Supplementary Fig. 6c). These data suggest that Acbp6 inhibition in the midgut potentially limits mitochondria-related metabolic activity, but induces ectopic lipid synthesis and lipid accumulation. To this end, we also uncovered that attenuating Acbp6 in enterocytes limits mitochondrial

**Fig. 2 | Nutrient-dependent metabolic network in the midgut is shaped by Acbp6 function. a** Diagram of select genes and metabolites involved in lipid synthesis and mitochondrial function. G3P, glycerol-3-phosphate; LPA, lysophosphatidic acid; PA, phosphatidic acid; DAG, diacylglycerol; TAG, triacylglycerol. **b**–**k** Nutrient-dependent changes in metabolic network upon enterocyte-specific depletion of Acbp6. **b** Neutral lipid/lipid droplet fluorescent staining of dissected posterior midguts during nutrient adaptation; stained with Bodipy (red, reduced state). **c** FASN immunostaining of dissected posterior midguts during nutrient adaptation, stained with anti-FASN (red) and DAPI (nuclei, blue). **d**–**e** Transcriptional changes (measured by qRT-PCR) of *Lipin* **d** and *Gpat* **e** in dissected whole midguts during nutrient adaptation. **d** From left to right, bars represent $n = 3, 3, 4, 3, 3$ and 3 independent samples. **e** $n = 3$ independent samples. **f** Mitochondrial ATP5A immunostaining of dissected posterior midguts during

nutrient adaptation, stained with anti-ATP5A (red) and DAPI (nuclei, blue). **g** Representative images of TMRE fluorescent histochemistry (red) of dissected posterior midguts during nutrient adaptation; TMRE (red). **h**–**k** Transcriptional changes (measured by qRT-PCR) of representative mitochondria genes, **h** *SdhA*, **i** *NdS1*, **j** *Mtpβ*, and **k** *Cpt1*, in whole dissected midguts during ad libitum (al) feeding, or fasting (F), or refeeding (RF). **h** From left to right, bars represent $n = 3, 3, 4, 3, 3$ and 4 independent samples. **i** $n = 3$ independent samples. **j** From left to right, bars represent $n = 3, 3, 3, 3, 4$ and 3 independent samples. **k** $n = 3$ independent samples. Bars represent mean ± SEM (unpaired 2-tailed Student's *t*-test). The exact *p* values are provided in figure. Source data are provided as a Source Data file. Fasting represents 2 days of nutrient deprivation, and Refeeding represents 2 days of refeeding. All genotypes are as follows; *w1118*; NP1Gal4/+ (controls, Ctrl.) and *w1118*; NP1Gal4/UAS-Acbp6 RNAi. Scale bars, 10 μm.

dynamics during nutrient adaptation. While control flies are able to maintain mitochondrial homeostasis in the midgut during fasting-refeeding transitions, NP1Gal4>UAS-Acbp6[RNAi] flies display a severe loss of mitochondria number and intensity (monitoring ATP synthase-α subunit [ATP5A] immunostaining, Fig. 2f, Supplementary Fig. 6d, f) and mitochondrial membrane potential/function (TMRE, Fig. 2g, Supplementary Fig. 6e, g), mainly in midgut enterocytes. This loss of mitochondrial homeostasis starts during fasting and is sustained during refeeding (2 days). We also leveraged mito-GFP (a human mitochondrial import sequence [hCox8A] fused to GFP) as a marker of mitochondrial number and density, and found in controls that mitochondrial numbers decrease during fasting and increase upon refeeding, reflecting the dynamics of mitochondrial adaptation to nutrient availability in the midgut (Supplementary Fig. 6h–i). Furthermore, inhibiting Acbp6 function in enterocytes (UAS-mito-GFP, NP1Gal4>UAS-Acbp6[RNAi]) impedes the recovery of mitochondrial numbers after refeeding, suggesting a critical role for Acbp6 in regulating mitochondrial number/density in response to nutrient fluctuations (Supplementary Fig. 6h–i). Acbp6 inhibition also leads to transcriptional downregulation of genes (in the midgut) required for mitochondrial electron transport chain (ETC) function and OXPHOS, as well as crucial beta-oxidation enzymes/substrate carriers (Fig. 2a, h–k). These data highlight the impact of Acbp6 function on mitochondrial dynamics and proper regulation of metabolic pathways in the posterior midgut during nutrient adaptation. Likely related to this metabolic adaptation, Acbp6 function in midgut enterocytes is both necessary and sufficient to dictate organismal sensitivity to starvation (Supplementary Fig. 6j–k).

We confirmed this aberrant 'switching' of metabolic networks (elevated lipid synthesis and attenuation of mitochondrial function) in *acbp6Δ1* mutant flies during fasting-refeeding transitions. Also, re-expressing Acbp6 (UAS-Acbp6) can rescue these phenotypes in *acbp6Δ1* heterozygote mutants, restoring mitochondrial numbers and function, as well as limiting lipid accumulation, in midgut enterocytes (Supplementary Fig. 7a–f, Supplementary Fig. 8a–e).

Our data suggest that Acbp6, induced in midgut enterocytes during nutrient deprivation when fatty acid activation and usage are enhanced, is required to partition acyl-CoA towards mitochondria, preserving mitochondrial homeostasis during nutrient adaptation (fasting-refeeding transitions).

## Acbp6 function tunes acetyl-CoA metabolism and protein acetylation to regulate midgut tissue plasticity during nutrition adaptation

We hypothesized that aberrant 'switching' of metabolic networks in the absence of Acbp6 is likely to influence more than just energy output from mitochondria during nutrient adaptation. Beta-oxidation of fatty acids (via acyl-CoA), which boosts acetyl-CoA and citrate levels to feed the TCA cycle, can sustain mitochondrial OXPHOS and energy production, primarily ATP (adenosine triphosphate) (Figs. 2a and 3a). Moreover, citrate can be exported from mitochondria into the

cytoplasm, where it freely diffuses into and out of the nucleus and can be converted into acetyl-CoA via ATP citrate lyase (ACLY)[15] (Fig. 3a). Through fatty acid activation/oxidation, acyl-CoA can presumably tune acetyl-CoA metabolism and nuclear/cytoplasmic acetyl-CoA pools. To this end, we found that ubiquitous over-expression of Acbp6 (DaGal4>UAS-Acbp6) can elevate not only ATP levels, but also boost citrate synthase activity and acetyl-CoA levels (Fig. 3b–d).

As previously mentioned, acetyl-CoA metabolism can dramatically influence cellular signaling and cellular transcriptional outputs through impacting acetylation of histones and non-histone proteins[21,57]. We uncovered that Acbp6-mediated changes in metabolism are required to promote general protein acetylation during nutrient adaptation in the midgut. In control flies, we found that general (pan-) lysine acetylation (an evolutionarily conserved post-translational modification) is strongly reduced in the midgut in response to nutrient deprivation (2 days fasting), but is restored 2 days after refeeding (Figs. 3e and 4e). Pan-lysine acetylation is revealed throughout midgut cell types, but is most prominent in enterocyte nuclei (minimally in enterocyte cytoplasm, Fig. 3e and Supplementary Fig. 9a). Conversely, attenuating Acbp6 in enterocytes utilizing RNAi (NP1Gal4>UAS-Acbp6[RNAi]) completely limits the restoration of pan-lysine acetylation only during refeeding (Figs. 3e, 4e, Supplementary Fig. 9b). Similar changes in midgut pan-lysine acetylation were found in *acbp6Δ1* heterozygote mutant flies during fasting-refeeding transitions (Supplementary Fig. 9c–e).

These data suggest, at least, that general protein acetylation in midgut enterocytes, shaped by Acbp6-dependent changes in acetyl-CoA metabolism, plays a role in nutrient-dependent tissue plasticity. Consequently, we first explored whether key enzymes that shape acetyl-CoA metabolism are required for Acbp6-dependent effects on midgut proliferative homeostasis during nutrient adaptation. We targeted both ACLY (ATP citrate lyase) and CPT1 (Carnitine palmitoyltransferase 1), a mitochondrial enzyme that promotes the movement of acyl carnitine (generated through acyl-CoA) from the cytosol into the mitochondria to drive beta-oxidation of fatty acids (Fig. 3a). Attenuating Acly or Cpt1, utilizing RNAi, in enterocytes that over-express Acbp6 (NP1Gal4>UAS-Acbp6, UAS-Acly[RNAi] or UAS-Cpt1[RNAi]) strongly inhibits midgut ISC activation during the transition from fasting to refeeding (2 days, monitored by increases in pH3+ cells and Dl+ cells, Fig. 3f, g and Supplementary Fig. 10a). While ACLY and CPT1 are required for genetically induced Acbp6-dependent (UAS-Acbp6) increases in ISC activation during refeeding, these enzymes are also generally required (compared to increases in control flies, Fig. 3f). Thus, Acbp6-mediated metabolic adaptation likely plays a pivotal role in shaping nutrient-dependent tissue plasticity by modulating proliferative homeostasis through the regulation of acetyl-CoA metabolism. This dynamic interplay allows for the fine-tuning of cellular processes and ensures proper adaptation to changing nutrient conditions. To this end, ACLY and CPT1 are also required for Acbp6-dependent changes in general protein acetylation in the midgut during fasting-refeeding transitions (Fig. 3h).

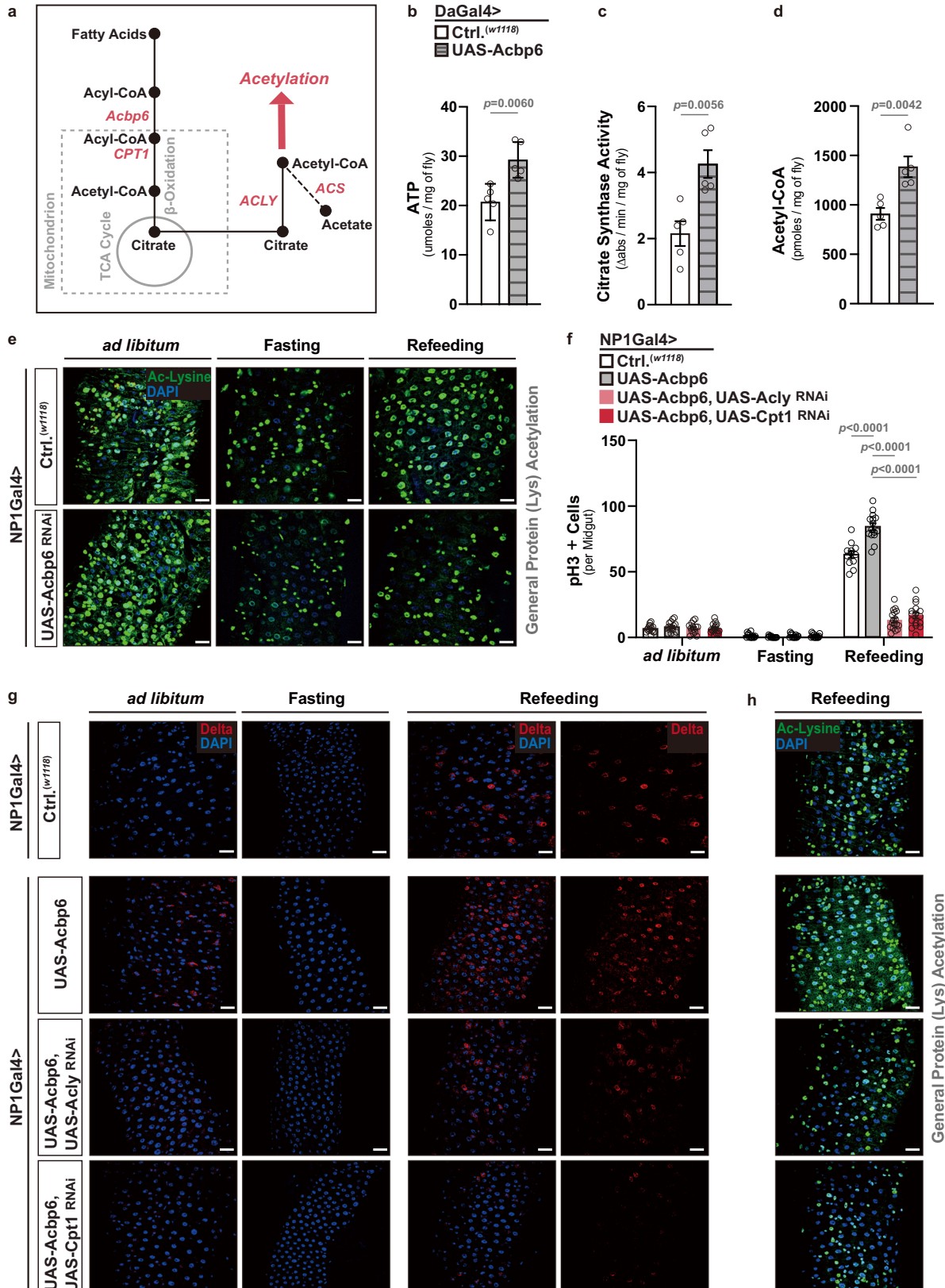

Next, we wanted to further substantiate the integration of Acbp6 function, acetyl-CoA metabolism, and acetylation in the midgut during nutrient adaptation. We therefore tested whether dietary acetate supplementation can bypass the requirement for Acbp6 in shaping metabolic network that influence acetyl-CoA metabolism. Independent of mitochondria, acetate can be directly converted to acetyl-CoA

through the enzyme Acetyl-CoA synthetase[25,58] (ACS, Fig. 3a), and we found that acetate supplementation within food can dose-dependently increase pan-lysine acetylation in the midgut during refeeding (in control flies, monitored by the number of acetylated lysine+ cells or acetylated lysine fluorescence intensity per area (within a section of the posterior midgut, Fig. 4a–c). Utilizing a non-saturating

**Fig. 3 | Acbp6 regulates acetyl-CoA metabolism and protein acetylation to adjust midgut proliferative homeostasis during nutrition adaptation.**
**a** Diagram of select genes and metabolites involved in acyl-CoA metabolism, acetyl-CoA metabolism, and acetylation. **b–d** Changes in various metabolite levels after whole-body upregulation of Acbp6 function. Genotypes; *w1118*; Daughterless (Da) Gal4/+ (controls, Ctrl.) and *w1118*; UAS-Acbp6/+; DaGal4/+. (5 whole flies per sample, *n* = 5 independent samples.) **b** Quantification of ATP (adenosine triphosphate) levels. **c** Quantification of citrate synthesis activity. **d** Quantification of acetyl-CoA levels. **e** Acetylated (Ac)-lysine immunostaining of dissected posterior midguts upon enterocyte-specific depletion of Acbp6 during nutrient adaptation, stained with anti-Ac-Lysine (green) and DAPI (nuclei, blue). Genotypes; *w1118*; NP1Gal4/+ (controls, Ctrl.) and *w1118*; NP1Gal4/UAS-Acbp6$^{RNAi}$. **f–h** Genetic requirement of ACLY and CPT1 in Acbp6-dependent control of proliferative homeostasis during nutrient adaptation. **f** Quantification of phospho-Histone (H3) positive cells (per

whole dissected midgut; from left to right, bars represent *n* = 15, 15, 14, 17, 15, 14, 15, 16, 12, 15, 17 and 16 independent samples), and immunostaining to detect **g** Delta positive cells in dissected posterior midguts (anti-Delta [red] and DAPI [nuclei; blue]), and **h** acetylated-lysine in dissected posterior midguts (stained with anti-Ac-Lysine (green) and DAPI (nuclei, blue) upon enterocyte-specific upregulation of Acbp6 or upregulation of Acbp6 and concurrent depletion of ACLY or CPT1 during nutrient adaptation (*n* = 3 independent experiments). Genotypes; *w1118*; NP1Gal4/+ (controls, Ctrl.), and *w1118*; NP1Gal4/UAS-Acbp6, and *w1118*; NP1Gal4, UAS-Acbp6/UAS-CPT1$^{RNAi}$, and *w1118*; NP1Gal4, UAS-Acbp6/UAS-ACLY$^{RNAi}$. Bars represent mean ± SEM (unpaired 2-tailed Student's *t*-test). The exact *p* values are provided in figure. Source data are provided as a Source Data file. Fasting represents 2 days of nutrient deprivation, and Refeeding represents 2 days of refeeding. Scale bars, 10 µm.

dose (50 mM) that doesn't impact pan-lysine acetylation in the midgut when flies are fed *ad libitum* or fasting (Fig. 4b, c, Supplementary Fig. 9b, Supplementary Fig. 10b), acetate supplementation can restore pan-lysine acetylation in NP1Gal4>UAS-Acbp6$^{RNAi}$ flies during refeeding (2 days, Fig. 4d, e, Supplementary Fig. 9b). Furthermore, acetate supplementation can also re-establish, in part, ISC activation/proliferation and restore midgut resizing during refeeding (2 days), different from *ad libitum* feeding or fasting, which is limited when Acbp6 is attenuated in midgut enterocytes (Fig. 4f, g, Supplementary Fig. 10c, d).

Taken together, these results highlight that Acbp6 and acyl-CoA metabolism in midgut enterocytes can dictate general (pan-) lysine acetylation, correlated with changes in ISC proliferative homeostasis, by tuning acetyl-CoA metabolism during nutrient adaptation. Distinctively, Acbp6 induction is thus linked with fatty acid activation during nutrient deprivation, providing acyl-CoA-dependent substrates for metabolic networks that presumably shape acetyl-CoA levels. This nutrient-dependent calibration of acetyl-CoA metabolism likely promotes rapid increases in general protein acetylation during refeeding, as well as tissue resizing, that dictate plasticity.

### Nutrient-dependent acyl-CoA metabolism targets STAT function to dictate midgut tissue plasticity

Subsequently, we wanted to more deeply investigate the mechanistic link between Acbp6-dependent changes in acetylation and proliferative homeostasis (i.e., activating intestinal stem cell proliferation from quiescence) during nutrient adaptation. We observed that Acbp6-dependent and histone-specific changes in lysine acetylation (as opposed to general protein pan-lysine acetylation, Fig. 3e and Supplementary Fig. 9a, c) are minimal during fasting-refeeding transitions, and are not robustly changed in large, polyploid enterocytes (monitored by pan-histone [H3-K9/K14/K18/K23/K27] lysine acetylation, Supplementary Fig. 11a). In totality, our data at least suggest that non-histone protein acetylation in midgut enterocytes, directed by acyl-CoA metabolism, is influencing cell signaling pathways and further impacting ISC activation during fasting-refeeding transitions. In order to identify such pathways, and in the absence of distinct molecular tools, we decided to perform biased functional genetic screening of signaling pathways that have been shown to both; (I) be directly or indirectly influenced by acetylation-dependent post-translation modifications, and (II) control ISC quiescence-activation (and vice versa) transitions. We selected 7 transcription factors to genetically target, including; STAT92e (the only STAT in *Drosophila*), Foxo, Relish (*Drosophila* NF-κB), HNF4, Nrf2, PGC1alpha, and Myc, as well as one receptor, Egfr. Utilizing RNAi, we found that attenuating STAT92e in midgut enterocytes (NP1Gal4>UAS-STAT92e$^{RNAi}$) could uniquely phenocopy Acbp6 loss-of-function effects on ISC activation, blocking both increases in ISC proliferation (Fig. 5a) and increases in stem cell number (Dl+ cells, Fig. 5b) during refeeding (2 days).

Signal transducers and activators of transcription 92e (STAT92e) is a transcription factor directed by the Jak/STAT signaling pathway[54].

This pathway has been shown to regulate critical homeostatic processes in germline and somatic stem cells in *Drosophila*, as well as regenerative processes in several tissues[54,59,60]. Furthermore, in mammals, acetylation of various members of the STAT family of genes is known to impact transcription factor activity, localization, and stability[61]. In control flies, we found that STAT92e activity is strongly enhanced in the midgut during fasting-refeeding transitions (using an in vivo 10XSTAT-GFP reporter, Fig. 5c). STAT-GFP reporter activity is maintained in stem/progenitor cells, but increased in large, polyploid enterocytes during refeeding (2 days, Fig. 5c and Supplementary Fig. 11b). However, attenuating Acbp6 in enterocytes (NP1Gal4>UAS-Acbp6$^{RNAi}$) restricts this nutrient-dependent STAT activation, indicating by the withdraw of STAT-GFP intensity in large, polyploid enterocytes (Fig. 5c). STAT92e activity is also sensitive to acetyl-CoA fluctuations, as acetate supplementation can further increase STAT-GFP reporter activity in enterocytes uniquely during refeeding, which correlates with elevated pan-lysine acetylation (Fig. 5d). These data suggest that STAT92e in midgut enterocytes is a crucial regulator downstream of Acbp6 and acyl-CoA-mediated acetyl-CoA metabolism that non-autonomously directs ISC proliferative homeostasis during nutrient adaptation.

### An Acbp6-STAT92e-cytokine signaling axis adjusts proliferative homeostasis during nutrient adaptation

We next wanted to identify mitogenic signals, released from enterocytes, that could non-autonomously regulate this nutrient-dependent ISC activation and tissue plasticity. We uncovered that an Unpaired (Upd) cytokine ligand, specifically Upd3, is required for shaping ISC proliferative homeostasis during nutrient adaptation. Upd cytokine ligands generated in enterocytes have been shown to be sufficient to stimulate ISC proliferation through activating the Jak/STAT signaling pathway in ISCs[54,62]. We found that *Upd3* transcription, but not *Upd1* or *Upd2*, is impaired during refeeding (2 days) when Acbp6 is attenuated in midgut enterocytes (NP1Gal4>UAS-Acbp6$^{RNAi}$, Fig. 6a). Additionally, *Upd3* transcription is also impaired when STAT92e is attenuated in midgut enterocytes (NP1Gal4>UAS-STAT92e$^{RNAi}$) in response to refeeding (2 days), whereas inhibiting Relish or Egfr in enterocytes (NP1Gal4>UAS-Relish$^{RNAi}$ or UAS-Egfr$^{RNAi}$) cannot repress *Upd3* transcription during fasting-refeeding cycle (Fig. 6b, Supplementary Fig. 11c). These data suggest that both Acbp6 and STAT92e are upstream of Upd3 regulation during nutrient adaptation. Furthermore, utilizing Upd3Gal4>UAS-GFP transgenic flies as an expression reporter, we observed that GFP induction and intensity is increased in midgut enterocytes uniquely during refeeding (2 days, Fig. 6c). Attenuating Acbp6 in this genetic background (i.e. in Upd3-expressing enterocytes; UAS-GFP, Udp3Gal4>UAS-Acbp6$^{RNAi}$) limits GFP induction during nutrient adaptation (Fig. 6c). Similar to STAT92e, Upd3 induction is also sensitive to acetyl-CoA fluctuations, as acetate supplementation can further increase this GFP induction during refeeding (Fig. 6d). These data suggest that STAT92e regulation of

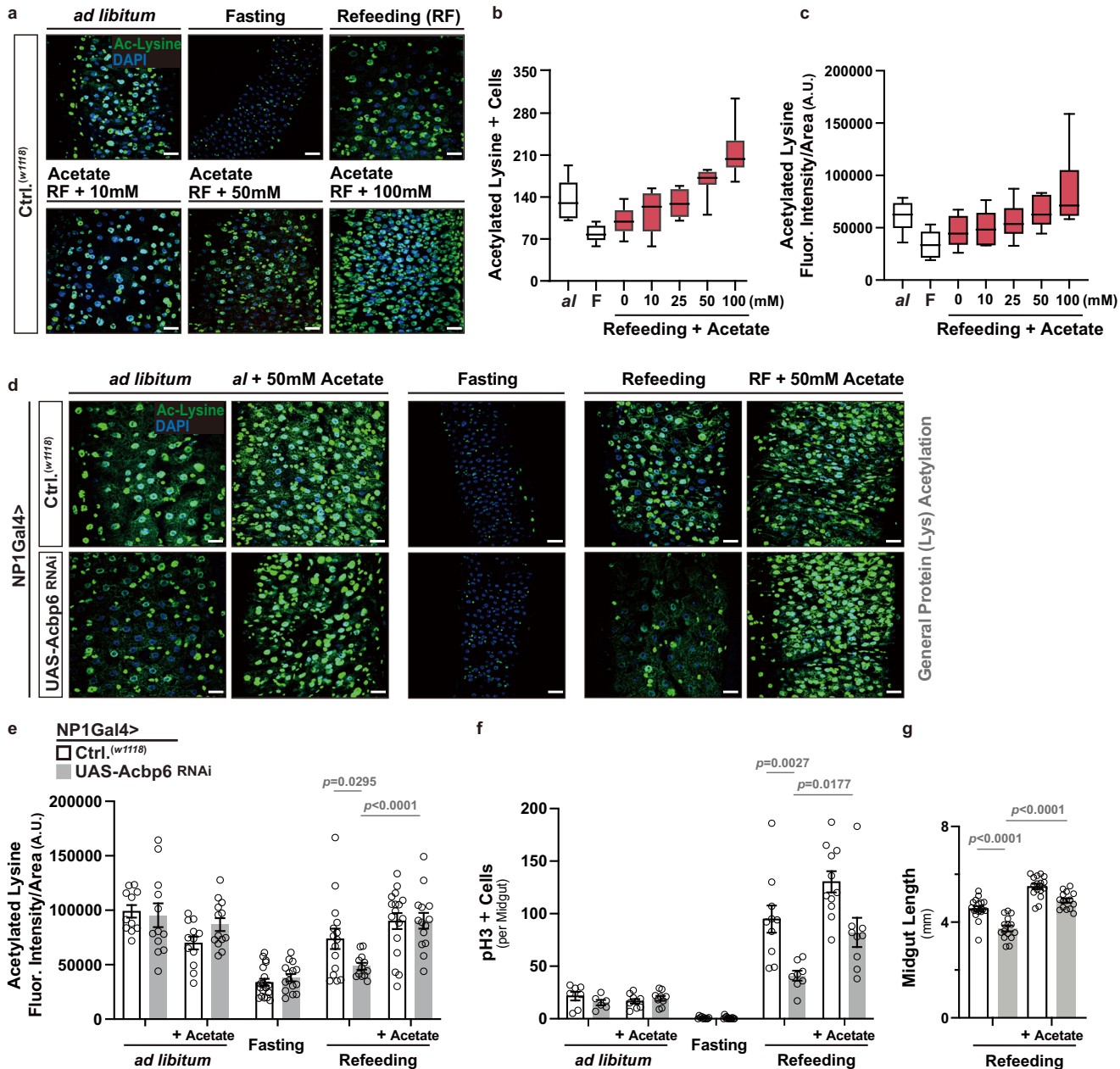

**Fig. 4 | Dietary acetate supplementation can bypass the requirement for Acbp6 in shaping acetyl-CoA metabolism and proliferative homeostasis. a–c** Effects of dietary acetate supplementation on pan-protein acetylation in the midgut during nutrient adaptation. **a** Acetylated (Ac)-lysine immunostaining of dissected posterior midguts in control flies (*w1118*; [controls, Ctrl.]) during nutrient adaptation and with dose-dependent dietary acetate supplementation, stained with anti-Ac-Lysine (green) and DAPI (nuclei, blue). Quantification of **b** acetylated-lysine positive cells in dissected posterior midguts (per field; *n* = 10 independent samples) and **c** acetylated-lysine fluorescence intensity/area within dissected posterior midguts (per field; A.U. [Arbitrary Units]; *n* = 10 independent samples) with dose-dependent dietary acetate supplementation during refeeding (RF). Data in **b, c** are presented as box plots, with the boxes representing the mean values (center line), and upper and lower quartiles values (box limits), and the whiskers indicating the range. **d** Acetylated (Ac)-lysine immunostaining of dissected posterior midguts upon enterocyte-specific depletion of Acbp6 during nutrient adaptation and with dietary acetate supplementation (50 mM) during *ad libitum* and refeeding conditions, stained with anti-Ac-Lysine (green) and DAPI (nuclei, blue). Genotypes; *w1118*; NP1Gal4/+ (controls, Ctrl.) and *w1118*; NP1Gal4/UAS-Acbp6 RNAi. **e–g** Quantification of **e** acetylated-lysine fluorescence intensity/area in dissected posterior midguts (per field; A.U. [Arbitrary Units]; from left to right, bars represent n = 11, 12, 12, 13, 19, 15, 15, 12, 17 and 14 independent samples), **f** phospho-Histone (H3) positive cells (per whole dissected midgut; from left to right, bars represent *n* = 7, 6, 9, 9, 10, 13, 10, 8, 11, 9, 11 and 12 independent samples), and **g** midgut length measurement during refeeding with dietary acetate supplementation (50 mM) (from left to right, bars represent *n* = 18, 15, 16 and 15). Genotypes; *w1118*; NP1Gal4/+ (controls, Ctrl.) and *w1118*; NP1Gal4/UAS-Acbp6[RNAi]. Bars represent mean ± SEM (unpaired 2-tailed Student's *t*-test). The exact *p* values are provided in figure. Source data are provided as a Source Data file. Fasting represents 2 days of nutrient deprivation, and Refeeding represents 2 days of refeeding. Scale bars, 10 µm.

Upd3 in enterocytes, ultimately driving Jak/STAT-mediated proliferation in intestinal stem cells, is required for nutrient-dependent changes in ISC proliferative homeostasis. To this end, attenuating STAT92e or Upd3, utilizing RNAi, in enterocytes that over-express Acbp6

(NP1Gal4>UAS-Acbp6, UAS-STAT92e[RNAi] or UAS-Upd3[RNAi]) strongly inhibits midgut ISC activation during the transition from fasting to refeeding (2 days, monitored by increases in pH3+ cells and Dl+ cells, Fig. 6e–g). Moreover, our findings demonstrate that the specific

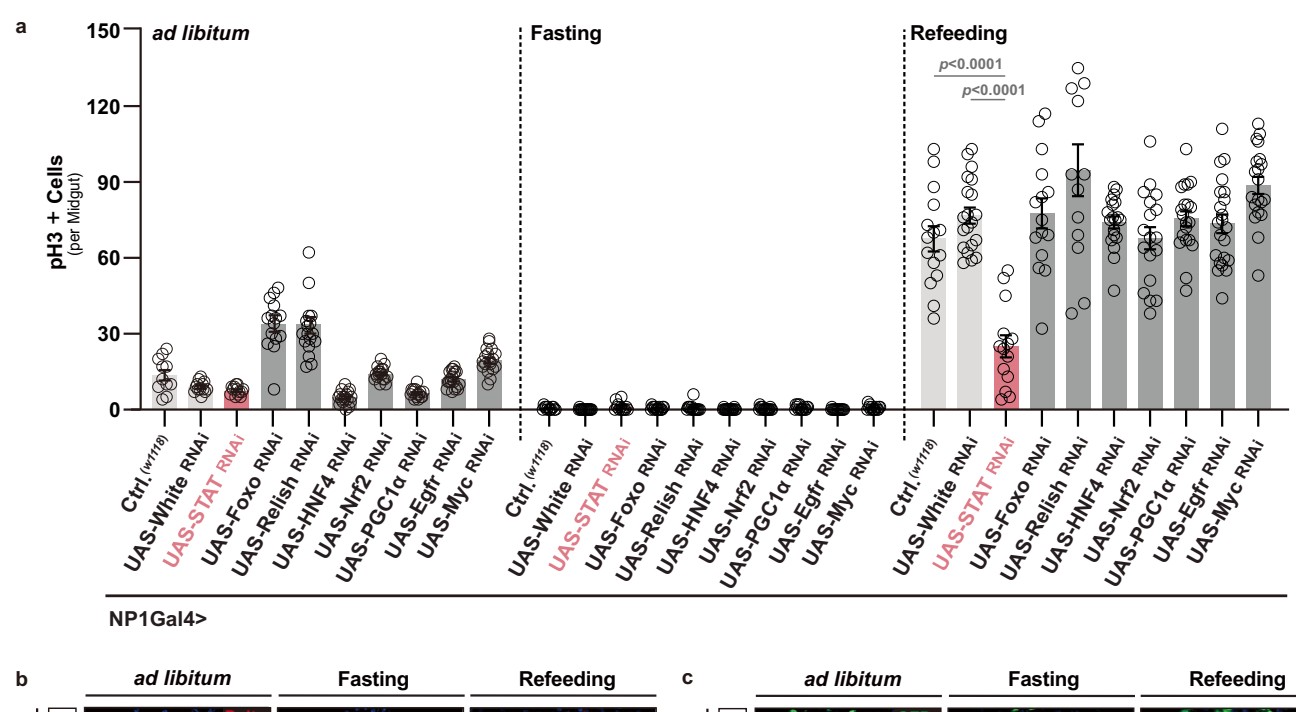

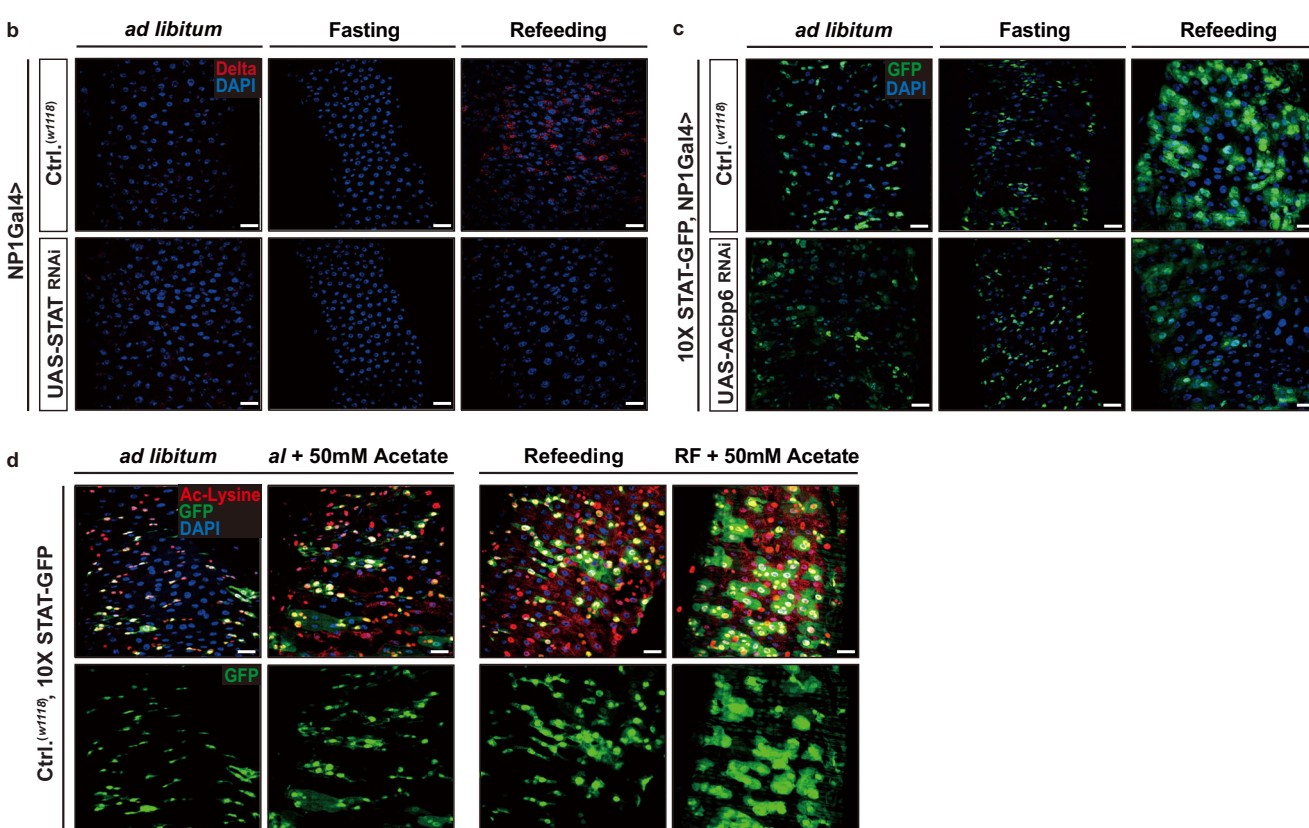

function of STAT92e in enterocytes is not essential for bacterial infection-induced midgut regenerative responses, as shown in Supplementary Fig. 11d–g. This observation suggests that enterocyte-derived STAT92e is not required to handle all external stresses but rather might play a slightly more specific role in nutrient adaptation processes.

These data highlight an Acbp6/acyl-CoA – STAT – Upd3 regulatory axis that shapes ISC proliferative homeostasis, tissue resizing, and plasticity during nutrient adaptation. Nutrient-dependent calibration of acetyl-CoA metabolism and general protein acetylation, tuned by Acbp6, also underlie this regulatory axis through influencing STAT activation in midgut enterocytes.

## Discussion

In totality, our findings revealed that Acbp6, which arose by evolutionary expansion in *Drosophila* with specialized expression in functionally differentiated midgut enterocytes, is a critical regulator of nutrient-dependent tissue remodeling. Our findings also add to the diversity of Acbp and Acbd (Acyl-CoA binding domain containing proteins) function across kingdoms, including physiological and pathophysiological conditions in humans, highlighting that acyl-CoA metabolic networks are evolutionarily conserved 'hubs' in metabolic control[63,64].

Nutrient availability is a critical determinant in animal (and population) survival, as nutritional deficiencies attenuate growth and

**Fig. 5 | Acbp6-mediated acetyl-CoA metabolism dictates STAT92e function to regulate midgut proliferative homeostasis during nutrient adaptation.**
**a** Biased functional genetic screening to identify key regulators of Acbp6-dependent changes in proliferative homeostasis during nutrient adaptation. Quantification of phospho-Histone (H3) positive cells (per whole dissected midgut) upon enterocyte-specific depletion of various transcription factors or receptors during nutrient adaptation (utilizing RNAi, from left to right, bars represent n = 11, 13, 13, 15, 18, 18, 17, 15, 19, 18, 13, 18, 13, 16, 15, 18, 18, 15, 15, 15, 20, 24, 14, 15, 13, 19, 18, 20, 22 and 20 independent samples). Genotypes; *w1118*; NP1Gal4/+ (controls, Ctrl.) and *w1118*; NP1Gal4/UAS-GeneX [RNAi]. White RNAi was used as an additional control (*w1118*; NP1Gal4/UAS-White [RNAi]). **b** Immunostaining to detect Delta positive cells in dissected posterior midguts upon enterocyte-specific depletion of STAT (STAT92e) during nutrient adaptation; anti-Delta (red) and DAPI (nuclei; blue) (n = 3 independent experiments). Genotypes; *w1118*; NP1Gal4/+ (controls, Ctrl.) and *w1118*;

NP1Gal4/UAS-STAT [RNAi]. **c** STAT92e activity (using 10X STAT-GFP transgenic reporter flies) in dissected posterior midguts during nutrient adaptation; stained with DAPI (nuclei, blue); GFP (green) (n = 3 independent experiments). Genotypes; *w1118*; 10X STAT-GFP, NP1Gal4/+ (controls, Ctrl.) and *w1118*; 10X STAT-GFP, NP1Gal4/UAS-Acbp6 RNAi. **d** Acetylated (Ac)-lysine immunostaining, and monitoring STAT92e activity, of dissected posterior midguts during nutrient adaptation and with dietary acetate supplementation (50 mM) during ad libitum and refeeding conditions, stained with anti-Ac-Lysine (green) and DAPI (nuclei, blue); GFP (green); *w1118*; 10X STAT-GFP/+ (controls, Ctrl.) flies (n = 3 independent experiments). Bars represent mean ± SEM (unpaired 2-tailed Student's t-test). The exact p values are provided in figure. Source data are provided as a Source Data file. Fasting represents 2 days of nutrient deprivation, and Refeeding represents 2 days of refeeding. Scale bars, 10 μm.

fecundity. Nutrients and the highly conserved nutrient-responsive Target of Rapamycin (TOR) signaling are also key factors in other somatic stress responses (such as viral infection), as mTOR activation can promote natural killer cell effector function to destroy virus-infected cells[65]. To this end, animals have evolved sophisticated molecular and metabolic mechanisms to adapt to complex environments. Metabolic flexibility is likely dependent, at least in a small part, on gene family expansion followed by expressional/functional divergence. Gene duplication (expansion) can facilitate metabolic flexibility through gene dosage, and neo- or sub-functionalization (tissue-specific functionalization)[66-68]. Previous studies have uncovered that around 23–25% of the gene families, after gene duplications, display tissue-specific expression partitioning or complementation of genes[69]. Similar to Decay-accelerating factors (DAFs) in mice or zinc finger with KRAB- and SCAN-domain proteins (ZKSCANs) in humans[69], *Acbps* represent a gene family in *Drosophila melanogaster* emerged by gene duplication events with a diversified expression pattern. This may reflect sub-functionalization of Acbps in some insect species during evolution to enhance metabolic flexibility, thus fulfilling unique energy requirements to environmental challenges.

Additionally, duplicated *Acbps* evolved two different expression patterns in *Drosophila melanogaster*. Four *Acbp* gene family members display expression in many tissues, while four other *Acbps* (including *Acbp6*) are specifically expressed in gastrointestinal tract. This likely highlights a heightened requirement for Acbps and acyl-CoA metabolism in the *Drosophila* midgut. Beyond nutrient absorption, the midgut and gastrointestinal tract also acts as a primary barrier epithelium that is further required to maintain systemic energy homeostasis[70]. Previous studies have revealed that the adult midgut can dynamically remodel its structure in response to oral infection, injury, or mating by adjusting epithelium renewal[54,62,71]. Further highlighting insect midgut plasticity, other studies have revealed that nutrient deprivation leads to midgut shrinkage through a variety of mechanisms[2,5,40], while refeeding can promote midgut regrowth by boosting intestinal stem cell number and activating asymmetric stem cell division though systemic endocrine signaling[2,40]. Here, we uncovered an acyl-CoA/acetyl-CoA-centric mechanism regulating midgut remodeling during nutrient adaptation, through mediating quiescence-activation transitions of intestinal stem cells. Our data suggest that Acbp- and nutrient-dependent calibration of acetyl-CoA metabolism is also crucial for tissue plasticity. Recently, *Drosophila* intestinal stem cells were revealed to autonomously use the hexosamine biosynthesis pathway to monitor nutrient levels (in coordination with hormonal insulin-signaling) and accordingly adjust division rates[72]. Thus, it is likely that coordination of enterocyte metabolic networks (such as those tuned by acyl-CoA/acetyl-CoA metabolism) and ISC metabolism (such as hexosamine biosynthesis) underly nutrient sensing and metabolic adaptation in tissue remodeling.

Besides glycolysis, lipids, through fatty acid oxidation, can act as a major source of acetyl-CoA used for histone acetylation and subsequent transcriptional regulation[33]. However, before fatty acid β-oxidation, lipids must be activated (as fatty acyl-CoAs) and transited into mitochondria. Acbp proteins, as intracellular acyl-CoA transporters, have been increasingly implicated in the trafficking and regulatory properties of fatty acyl-CoA[38,73]. Here, we found that *Drosophila* Acbp6 is required to maintain a balance between lipogenesis and mitochondrial function, and Acbp6 depletion, which will impair acyl-CoA flux into mitochondria, likely leads to attenuation of acetyl-CoA production. Similar Acbp6-dependent changes metabolic pathways are observed in human gliomagenesis, as Acbp depletion blocks fatty acid oxidation and mitochondrial metabolic rate in glioblastoma cells[38]. In addition, depletion of Acb1 (ACBP in the fission yeast *Schizosaccharomyces pombe*) can lead to mitochondrial fragmentation, impaired oxygen consumption rate and obstructed cell proliferation[74]. Our work, together with other previously published studies, support an evolutionarily conserved and integral role for Acbps in the regulation of mitochondrial homeostasis.

Acyl-CoA- and mitochondria-dependent metabolic networks are also essential for generating acetyl-CoA pools, and acetyl-CoA is a rate-limiting metabolite for histone and non-histone protein acetylation[20,21]. Our data define a role for Acbp6 in tuning acetyl-CoA metabolism and protein acetylation, and we further identified STAT92e (the only STAT in *Drosophila*) as signaling node impacted by Acbp6-mediated acetyl-CoA metabolism (directly or indirectly). Numerous studies have revealed that STAT transcription factors play a key role in the signal transduction of cytokines and growth factors, and acetylation of STAT1, STAT2, STAT3, STAT5b and STAT6 in mammals has been shown to induce transcriptional activation and enhance DNA binding affinity[61,75]. In *Drosophila*, the JAK/STAT signaling pathway has been intensively studied as a key regulator of midgut regeneration and proliferative homeostasis. Infection or tissue damage can induce secretion of cytokines (Upd2 and Upd3) in enterocytes and enteroblasts that can activate JAK/STAT signaling in intestinal stem cells to drive proliferation[54,62]. However, we found that STAT activation in enterocytes, induced by Acbp6-dependent changes in acyl-CoA/acetyl-CoA metabolic processes, is also required to adjust intestinal stem cell proliferative homeostasis. This suggests that STAT92e may be directly acetylated by Acbp6-dependent acetyl-CoA abundance and sequential acetylation to transcriptionally regulate Upd3 expression and secretion. This regulatory mechanism seems distinct in nutrient-dependent tissue remodeling, as it is dispensable during pathogenic infection, and potentially underlies differential metabolic control of STAT activation that could broaden function of the single STAT gene in *Drosophila*. In addition to influencing JAK/STAT signaling, our small-scale meta-analysis also suggests that Acbp6 may potentially have a role in chronic stress conditions, such as aging. JAK/STAT signaling has previously been implicated in *Drosophila* midgut aging and mortality through

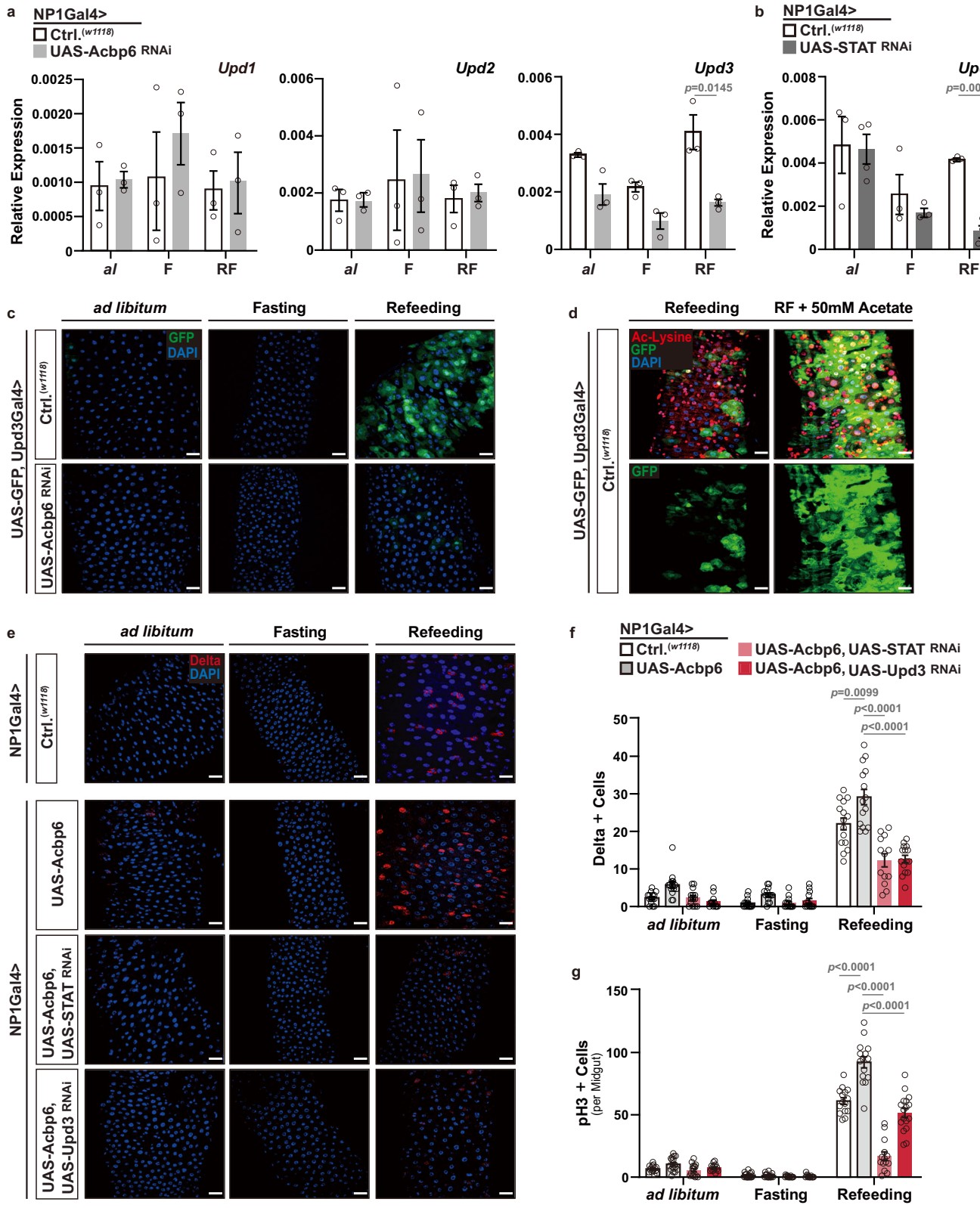

controlling aspects of midgut regeneration[76], suggesting that nutrient-dependent control of proliferative homeostasis through acyl-CoA binding proteins may also influence tissue plasticity during aging.

Overall, our data thus define a divergent regulatory mechanism, shaped by Acbp6-mediated acyl-CoA metabolism, that is required to adjust stem cell quiescence and activation, and therefore drive tissue plasticity during nutrient adaptation.

## Methods

### *Drosophila* husbandry and strains

The following strains were obtained from the Bloomington *Drosophila* Stock Center: *w1118* (#3605), Daughterless (Da)-Gal4 (#55851), UAS-nuclear localization sequence (nls)-GFP (#4775), UAS-GFP (derived from #39760), and Tubulin (tub)-Gal80 (temperature sensitive)-ts (#65406), UAS-mito-HA-GFP (#8443), UAS-Acbp3^RNAi (#58343). The

**Fig. 6 | Acbp6 can adjust midgut proliferative homeostasis through a STAT92e-Upd3 signaling axis during nutrient adaptation. a** Transcriptional changes (measured by qRT-PCR) of *Upd1*, *Upd2* and *Upd3* upon enterocytes-specific Acbp6 depletion during nutrient adaptation (*n* = 3 independent samples). Genotypes; *w1118*; NP1Gal4/+ (controls, Ctrl.) and *w1118*; NP1Gal4/UAS-Acbp6^RNAi. **b** Transcriptional changes (measured by qRT-PCR) of *Upd3* upon enterocyte-specific STAT (STAT92e) depletion during nutrient adaptation (from left to right, bars represent *n* = 3, 4, 3, 3, 3, and 4 independent samples). Genotypes; *w1118*; NP1Gal4/+ (controls, Ctrl.) and *w1118*; NP1Gal4/UAS-STAT^RNAi. **c** Upd3 expression patterns (using Upd3Gal4/UAS-GFP transgenic flies) in dissected posterior midguts during nutrient adaptation; stained with DAPI (nuclei, blue); GFP (green) (*n* = 3 independent experiments). Genotypes; *w1118*; UAS-GFP, Upd3Gal4/+ (controls, Ctrl.) and *w1118*; UAS-GFP, Upd3Gal4/UAS-Acbp6^RNAi. **d** Acetylated (Ac)-lysine immunostaining, and monitoring Upd3 induction, of dissected posterior midguts during refeeding and with dietary acetate supplementation (50 mM), stained with anti-Ac-Lysine (green) and DAPI (nuclei, blue); GFP (green); (*n* = 3 independent experiments); *w1118*; UAS-GFP, Upd3Gal4/+ (controls, Ctrl.) flies. **e–g** Genetic requirement of STAT92e and Upd3 in Acbp6-dependent control of proliferative homeostasis during nutrient adaptation. **e** Immunostaining of Delta positive cells in dissected posterior midguts (anti-Delta [red] and DAPI [nuclei]; blue), and quantification of **f** Delta positive cells in dissected posterior midguts (per field; from left to right, bars represent *n* = 15, 14, 14, 12, 14, 14, 14, 13, 15, 15, 13 and 15 independent samples) and **g** phospho-Histone (H3) positive cells (per whole dissected midgut; from left to right, bars represent *n* = 14, 17, 15, 14, 22, 18, 15, 15, 15, 15, 14, and 17 independent samples) upon enterocyte-specific upregulation of Acbp6 or upregulation of Acbp6 and concurrent depletion of STAT or Upd3 during nutrient adaptation. Genotypes; *w1118*; NP1Gal4/+ (controls, Ctrl.), and *w1118*; NP1Gal4/UAS-Acbp6, and *w1118*; NP1Gal4, UAS-Acbp6/UAS-STAT^RNAi, and *w1118*; NP1Gal4, UAS-Acbp6/UAS-Upd3 ^RNAi. Bars represent mean ± SEM (unpaired 2-tailed Student's t-test). The exact *p* values are provided in figure. Source data are provided as a Source Data file. Fasting (F) represents 2 days of nutrient deprivation, and Refeeding (RF) represents 2 days of refeeding. *Ad libitum; al.* Scale bars, 10 µm.

following strains were obtained from Vienna *Drosophila* RNAi Center: UAS-Acbp6^RNAi (#104642), UAS-Acbp4^RNAi (#109198), UAS-Acbp5^RNAi (#23586), UAS-Acly^RNAi (#30282); UAS-CPT1^RNAi (#4046); UAS-Schlank^RNAi (#33897), UAS-STAT92e^RNAi (#43866); UAS-Foxo^RNAi (#106097); UAS-Relish^RNAi (#49413); UAS-HNF4^RNAi (#12692); UAS-Nrf2^RNAi (#37673); UAS-PGC1α^RNAi (#103355), UAS-Egfr^RNAi (#107130), UAS-dMyc^RNAi (#2947), and UAS-White^RNAi (#30033). esgGal4 was kindly provided by H. Jasper. NP1Gal4 was kindly provided by D. Ferrandon, and 10xSTAT-GFP was kindly provided by E. Bach. Upd3Gal4 and UAS-Upd3^RNAi was kindly provided by N.Perrimon.

UAS-Acbp6 transgenic flies, Acbp6-RFP transgenic reporter flies, Acbp6^P-ACBP6^V5 transgenic flies, and *Acbp6Δ1* transgenic mutant flies were generated for this study.

All flies were reared on a standard yeast and cornmeal-based diet at 25 °C and 65% humidity on a 12 hr light and dark cycle, unless otherwise indicated. The standard lab diet (cornmeal-based) was made with the following protocol: 14 g Agar, 165.4 g Malt Extract, 41.4 g Dry yeast, 78.2 g Cornmeal, 4.7 ml propionic acid, 3 g Methyl 4-Hydroxybenzoate and 1.5 L water. To standardize metabolic results, fifty virgins were crossed to 10 males and kept in bottles for 2-3 days to lay enough eggs. Progeny of crosses were collected for 3-4 days after initial eclosion. Collected progeny (males and females) were then transferred to new bottles, and allowed them to mate for 3 days (representing unique populations fed *ad libitum*). All these flies were reared on a standard lab diet at 25 °C and 65% humidity on a 12 hr light and dark cycle, unless otherwise indicated. Next, post-mated females (20 flies per vial/cohort) were sorted into individual vials for nutrient deprivation (starvation media containing 9 mg Agar and 3ul propionic per 1 mL water – used to maintain midgut integrity). After 2 days of starvation, fly cohorts were transferred into vials containing a standard lab diet (refeeding).

For dietary supplementation of acetate, diluted sodium acetate (Sigma, #S2889) was incorporated into the standard lab diet or starvation media at various concentrations, resulting in 10 mM, 25 mM, 50 mM, and 100 mM acetate concentration within food.

Post-mated female flies were used for all experiments, because of sex-specific differences in midgut regeneration.

Transgenic lines were backcrossed 10 times into the *w1118* background that was used as a control strain, with continued backcrossing every 6-8 months to maintain isogeneity. All experimental genotypes were assayed for developmental defects (developmental timing, growth, fly size, and organ size), and no gross anatomical deficiencies were noted in any genotype represented in the results.

### Generation of transgenic flies

UAS-Acbp6 transgenic flies were generated by PCR amplification of the Acbp6 coding sequence from adult *Drosophila* (*w1118*) cDNA, with specific primers (F: TAAGCGGCCGCATGCCCACCGTAAGTCTTAC, R:

TAAGGTACCTTATTCGTACTGGGGAGCGT), and then cloned into a pUASt plasmid[77]. This plasmid was injected into *w1118* embryos (Rainbow Transgenic Flies).

Acbp6-RFP transgenic reporter flies were generated by PCR amplification of an upstream *Acbp6* promoter/enhancer sequence with specific primers (F: TAAGCATGCGAGGCAGCCTTCTAAGCGA, R: TAACTCGAGTTGGCAAACATGGCTCACA) from *w1118* genomic DNA, and cloned into a pB-RFP plasmid[78]. pB-Acbp6-RFP was injected into *Drosophila w1118*; attp40 embryos with a phiC31 integrase helper plasmid (Rainbow Transgenic Flies).

Acbp6^P-ACBP6^V5 transgenic flies were generated by utilizing the pB-Acbp6-RFP plasmid (described above) and replacing the RFP sequence with an *Acbp6* cDNA sequence tagged with V5-His, generated by gBlock:IDT. The new plasmid pB-Acbp6^P-ACBP6^V5 was injected into *Drosophila w1118*; attp40 embryos with a phiC31 integrase helper plasmid (Rainbow Transgenic Flies). The Acbp6^V5-His gBlock sequence is as follows: TAAACCGGTATGCCCACCTTTGAGGAGATCGTCGAGAAGG CCAAGAACTTCAAGAACCTGCCTAGCAAGGAGGAGTTCCTCGAGTTC TACGGCTACTACAAGCAGGCCACCGTCGGCGACTGCAACATCGAGGA GCCGGAAGATGAGGAGAAGAAGGCCCGCTACAACGCCTGGAAGAGC AAGGCCGGTCTGACCGCCGATGATGCCAAGGCCTACTACATCGAG GTCTACAAGAAGTACGCTCCCCAGTACGAATTAGAGAGCCGAGGGC CCTTCGAAGGTAAGCCTATCCCTAACCCTCTCCTCGGTCTCGATTCT ACGCGTACTGGGCATCATCACCATCACCATTGATCTAGAAAT.

*acbp6Δ1* mutant *Drosophila* were generated by using CRISPR-Cas9. Briefly, a transgenic fly line expressing Cas9 protein using the ubiquitous promoter actin (BDSC: #79743) was crossed to a second fly line expressing a custom *acbp6* guide RNA (gRNA sequence: CTTCAAGAACCTGCCTAGCAAGG). The genetic cross produces offspring with an active Cas9–gRNA complex, which cleaves and mutates the target site. *acbp6* mutants were sequenced, and the *acbp6Δ1* mutant was identified and characterized. *acbp6Δ1* mutants were verified by using PCR and electrophoresis with specific primers (wildtype/mutant-F: GAGGCAGCCTTCTAAGCGA, wildtype-R: CTCCTTGCTAGG-CAGGTTC, *acbp6Δ1*-R: ACTCCTCCTGGCAGGTTC).

### Conditional expression of UAS-linked transgenes

The TARGET system was used in combination with NP1Gal4 to conditionally express UAS-linked transgenes in enterocytes (NP1Gal4, tub-Gal80^ts)[79]. Flies were developed at 18 °C then shifted to 29 °C to induce transgene expression post-eclosion (Day 5).

### Transcriptomic meta-analysis

Previously generated transcriptomic datasets were used for a meta-analysis to uncover over-lapping genes/gene networks. These datasets include responses to septic infection/injury or fungal infection, viral infection, enteric infection, microbiota changes, and aging in *Drosophila*[44–48]. Annotated and analyzed datasets, as presented in the

referenced manuscripts, were utilized. Significantly upregulated genes (with a cut-off at ≥2-fold upregulated) were selected from each dataset to use in the analysis, and up-regulated genes common to more than one group were analyzed and compared by using ImageGP[80].

## Phylogenetic analysis

Using Acbp protein sequences as query sequences, a comprehensive search of Acbp homologues in insects was performed in NCBI, Ensembl, and FlyBase by TBLASTN. After removing the partial sequences and redundant sequences, the final data set included 85 complete Acbp protein sequences. Multiple sequence alignments of these homologues were conducted with MAFFT[81], and a Maximum-likelihood (ML) tree was reconstructed with PhyML (http://www.phylogeny.fr/). A phylogenetic tree was visualized using Interactive Tree of Life (iTOL) software (http://itol.embl.de).

## Analysis of gene expression

The transcriptional pattern of Acbp family members in adult flies was investigated using FlyAtlas2 Anatomy RNA-Seq data[82]. Expression was recorded as TPKM (transcripts per kbp per million reads) followed by Log2 transformation. Expression-based heat maps were performed using Heatmapper (http://www.heatmapper.ca/expression/).

Total RNA from intact fly intestines, dissected posterior midgut or whole flies was extracted using Trizol and complementary DNA synthesized using Superscript III (Invitrogen). Real-time PCR was performed using SYBR Green, the Thermo Fisher QuantStudio 5 Real Time PCR system, and the primers pairs described in the extended experimental procedures (Table S1). Results are average ± standard error of at least three independent samples, and quantification of gene expression levels calculated using the △Ct method and normalized to actin5C expression levels (plotted as relative expression).

## Comparative analysis of promoter/enhancer regions of Acbp genes

Upstream regulatory elements (approximately 1,000 bp) encompassing putative promoter/enhancer regions of various Acbp genes (Acbp3/4/5/6) was employed to determine the upstream transcriptional regulators influencing Acbp6. Determination of putative transcription factor (TF) binding sites, and thus transcription factors, associated with specific Acbp gene promoter/enhancer regions was achieved utilizing the Jaspar TF database, accessible through the UCSC Genome Browser (genome.ucsc.edu). Through the identification of potential TF binding sites, distinct regulatory factors governing the expression of Acbp6 were ascertained.

## Immunostaining and microscopy

Fly midguts were dissected in 1xPBS and fixed for 30 min at room temperature in gut fixation solution (100 mM Glutamic Acid, 25 mM KCl, 20 mM MgSO₄, 4 mM Na2HPO4, 1 mM MgCl₂; pH adjusted to 7.5 with 10 N NaOH, 4% Formaldehyde). Subsequently, all incubations were in PBS, 0.5% BSA, and 0.1% Triton-X at 4 °C. All primary antibodies were applied overnight. The following primary antibodies were used: rabbit anti-Acetylated lysine (Cell Signaling, #9441, 1:500), rabbit anti-Acetyl-Histone H3-K9/K14/K18/K23/K27 (ABclonal, #A17917, 1:500), rabbit anti-ATP5A (Abcam, #14748, 1:500), rabbit anti-V5 (Sigma V8137), rabbit anti-phospho-Histone 3 (Cell Signaling, #9701, 1:500), mouse anti-FASN1 (Dev. Studies Hybridoma Bank, 1:10), and mouse anti-Delta (Dev. Studies Hybridoma Bank, 1:10). Fluorescent secondary antibodies (Jackson Immunoresearch, 1:500) and 40,6-diamidino-2-phenylindole (DAPI, 1:500) used to stain DNA was incubated at room temperature for 2 h and 30 min, respectively. Confocal images were collected using a Nikon Eclipse Ti confocal system (utilizing a single focal plane) and processed using the Nikon software and Adobe Photoshop. For quantitative analysis, images were analyzed using ImageJ/Fiji software (https://imagej.net/software/fiji/).

## Bodipy staining and microscopy

Fly guts were dissected in PBS, then fixed in 4% formaldehyde/PBS for 20 min at room temperature. After washing with PBS for 10 min (3 times), guts were incubated in 2 µM Bodipy (Invitrogen #D3861) solution for 1 h at room temperature. Then washed with PBS for 10 min (3 times), and incubated in DAPI for 30 min. Guts were washed with PBS for 10 min (3 times), then mounted with Mowiol media. Samples were imaged by using Nikon Eclipse Ti confocal system (utilizing a single focal plane).

## Midgut length measurements

Drosophila midgut lengths were obtained by taking bright-field images using a Leica M165 FluoCombi stereoscope system and processed using Leica software. Measurements were made from proventriculus to posterior midgut-hindgut junction.

## TMRE (mitochondrial membrane potential) staining and microscopy

For midgut TMRE (tetramethylrhodamine, ethyl ester) staining, dissected in 1×PBS and fixed for 10 min at room temperature in gut fixation solution (100 mM Glutamic Acid, 25 mM KCl, 20 mM MgSO₄, 4 mM Na₂HPO₄, 1 mM MgCl₂; pH adjusted to 7.5 with 10 N NaOH, 4% Formaldehyde). Then incubated in 200 nm TMRE staining solution (Abcam, #113852) for 20 min at room temperature. After staining, samples were then rinsed once in wash solution (25 nm TMRE) for 30 s. Samples were quickly mounted in the wash solution onto a slide, kept overnight at 4 °C and imaged using identical setting on the confocal microscope. Confocal images were collected using a Nikon Eclipse Ti confocal system (utilizing a single focal plane) and processed using the Nikon software and Adobe Photoshop.

## Metabolite measurements

ATP measurement was conducted by using an ATP Determination Kit (Invitrogen, #A22066)[83]. Briefly, five adult flies (without head, per sample) were homogenized by using pellet pestle mixer on ice for 1 min in 100 µl of homogenization buffer (6 M guanidine HCl, 100 mM Tris (pH 7.8), 4 mM EDTA). Then samples were boiled for 5 min and centrifuged for 3 min at 20,000 g (4 °C). Ten microliters of supernatant were transferred into a 1.5 ml Epp tube and diluted 1:75 by adding dilution buffer, then added 10 µl of diluted supernatant to the individual wells of a 96 well plate. One hundred microliters of luciferase reaction mix were added to each well, and luminescence was immediately determined by Multimode Plate Reader (Perkin Elmer). ATP levels were normalized to weight.

Acetyl-CoA levels were measured by using an Acetyl-Coenzyme A Assay Kit (Sigma-Aldrich, #MAK039) according to the manufacturer's instructions. Briefly, five adult flies (without head) were homogenized by using pellet pestle mixer on ice for 1 min in 100 µl of ice-cold Acetyl-CoA Assay Buffer. Samples were centrifuged at 13,000 g for 10 min at 4 °C, then the supernatant was used to do the measurement. Ten microliters of the supernatant were brought to 50 ml per well with Acetyl-CoA Assay Buffer. Reaction mix was added to each well of 96-well plate, then measured with fluorescent spectrometry (530 nm for excitation and 580 nm for emission) by using Multimode Plate Reader (Perkin Elmer). Acetyl-CoA levels were normalized to weight.

## Citrate synthase activity assay

Five adult flies (without head) were homogenized by using pellet pestle mixer on ice for 1 min in 100 µl of extraction buffer (20 mM HEPES [pH 7.2], 0.1% Triton X-100, and 1 mM EDTA), then 400 µl extraction buffer were added to each sample. CS activity was measured by using 1 µl diluted cell lysate in a total of 150 µl reaction solution (0.1 mM of dithio-bis-nitrobenzoic acid [DTNB, Sigma-Aldrich], 0.3 mM acetyl-CoA, 1 mM oxaloacetic acid [Sigma-Aldrich], and 50 mM Tris-HCl [pH 8.0]). Absorbance was measured at 412 nm every 30 s for 30 min at

25 °C by using Epoch plate reader (BioTek). CS activity was normalized to weight.

## Food intake and feeding measurements

Twenty flies were transferred to vials filled with identical medium containing 0.5% brilliant blue (FD&C blue dye #1). Feeding was interrupted and 5 flies each were transferred to 200 ml 1× PBS containing 0.1% Triton X-100 (PBST) and homogenized immediately for 1 min by using pellet pestle mixer. Then samples were centrifuged at 4000 rpm and remove the pellet. Blue dye consumption was quantified by measuring absorbance of the supernatant at 630 nm (A630) by using Epoch plate reader (BioTek).

## Starvation sensitivity assay

Adult flies (20 flies per vial/cohort) were put into individual vials for nutrient deprivation (starvation media containing 9 mg Agar and 3ul propionic per 1 mL water – used to maintain midgut integrity), and flies were flipped into new vials every 2 days. The number of dead flies in each vial was recorded every 24 h, and data is presented as the mean survival of cohorts.

## Eclosion rate analysis

A total of 20 virgin flies were mated with ten males and placed in vials for 3 days to ensure sufficient egg laying. Following this, the parent flies were removed from the vials. The resulting progeny were allowed to develop into the pupal and adult stages, and the number of pupae and adult flies were counted to calculate the eclosion rate, representing the proportion of flies that successfully emerged from the pupal stage into adulthood.

## Oral infection assays

*Pseudomonas entompophila* (P.e.) was used for natural (oral) infections. Briefly, for oral infection, flies (7–10 d of age) were placed in a fly vial with food/bacteria solution and maintained at 25 °C. The food solution was obtained by mixing a pellet of an overnight culture of bacteria ($OD_{600} = 50$) with a solution of 5% sucrose (50/50) and added to a filter disk that completely covered the surface of standard fly medium. Midguts of infected flies were dissected 20 h after oral contact with infected food. At least 3 vials (cohorts of 15 flies per vial) were used for each genotype for subsequent analysis (immunostaining and qRT-PCR).

## Quantification and statistical analysis

Samples sizes were predetermined by the requirement for statistical analysis, and at least 3 biological replicates were utilized for all experiments. For all quantifications, n represents the number of independent samples/biological replicates, and error bar represents SEM. Statistical significance was determined using the unpaired *Student's t*-test in GraphPad Prism Software, and expressed as *P* values. Exact values of all n's can be found in Figure legends and individual data points are represented in all histograms.

## Reporting summary

Further information on research design is available in the Nature Portfolio Reporting Summary linked to this article.

## Data availability

All relevant data supporting the findings of this study are available within the article and its Supplementary Information/ Source Data files. Published gene expression datasets used in this study can be find from NCBI's Gene Expression Omnibus GSE36582 and GSE42726, Proc. Natl. Acad. Sci. USA, 10.1073/pnas.261573998, Cell Host Microbe, 10.1016/j.chom.2009.01.003, and Mbio, 10.1128/mbio.01117-14[44–48]. Public databases used in this study: FlyBase (https://flybase.org), FLYATLAS 2

(flyatlas2.org), JASPER TF database (https://jaspar.genereg.net), Ensembl (https://ensemblgenomes.org/), NCBI (https://www.ncbi.nlm.nih.gov/). Source data are provided with this paper.

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

## Acknowledgements
This work was supported by the National Institute of Diabetes and Digestive and Kidney Diseases (grant R01 DK133294 and grant R56 DK108930 to J.K.).

## Author contributions
X.L. designed and performed experiments, as well as wrote the manuscript. J.K. designed experiments and wrote the manuscript.

## Competing interests
The authors declare no competing interests.
