## [Peer Review File · Nature Communications]

REVIEWER COMMENTS

Reviewer #1 (Remarks to the Author):

This study by Li & Karpac presents a significant work on the role of Acbp6 in starvation induced tissue plasticity and acyl-CoA metabolism. The authors show right in the beginning that Acbp6 is a crucial regulator of midgut remodeling, including proliferation and acyl-CoA sequestering upon refeeding. They also show that Stat92e (and further Upd3) is acting directly or indirectly downstream of Acbp6 and is indispensable for Acbp6 regulated tissue plasticity during starvation. The experiments are well designed and performed, and the authors have utilized the many of the strengths Drosophila model has to unravel the role of Acbp6 in tissue remodelling and nutrient utilization upon starvation. The manuscript is well-written and presents an important piece of work for this field. Along with some issues the authors should correct/improve, this study could be strengthened by a few additional experiments suggested below.

Specific comments:

Row 122: How the authors have reasoned the study in the beginning is illogical. The authors state that they have “employed a small-scale meta-analysis of accessible *D. melanogaster* transcriptomes” and that these transcriptomes include ones that show changes to external cues including nutrition. However, none of the references include “nutritional” transcriptomes, but 3 of infection, 1 of microbiota and 1 of aging. Considering, that the whole focus of the study is on a nutrition-responsive gene Acbp6, the authors should include or at least discuss the existing nutritional transcriptomes and whether Acbp6 has been already shown to be regulated by nutritional cues. The reasoning between (largely) viral transcriptomes and a study of tissue plasticity during nutrition adaptation would need more clarification.

Many immunostainings lack quantification. Please add them with all figures with immunostainings. I acknowledge that this may be difficult for some cases and in these situations adding a panel of images to supplement could be an option.

While the role of Acbp6 in regulating tissue plasticity and acetyl-CoA metabolism during starvation-refeeding is very clearly shown by the authors, a big question remains on the role of Acbp6 in prolonged starvation. To clarify this profound question, the authors could include data on the starvation sensitivity of flies upon loss / overexpression of Acbp6 in the midgut (and also Acbp6 null mutants).

As the authors state on row 146 that Acbp6 is “specifically” a key regulatory node in nutrient adaptation. To make such a conclusion, the authors should show how other Acbp3, 4 and 5 are expressed compared to Acbp6, and if they are required in the midgut for tissue plasticity during starvation-refeeding in a similar manner to Acbp6. Moreover, this conclusion starting on row 145 is not quite in line with the results shown by this far (only the phylogenetic comparison and expression data are shown thus far) and should be corrected accordingly.

Row 154: It would be important to note in the beginning that Acbp6 is also expressed at high levels in the middle midgut. To clarify this, the authors could show the expression levels of Acbp6 +/- starvation + RF, in different parts of the gut with a horizontal diagram of the fly gut.

Row 242: The authors show that FASN protein levels (and later Lipin and Gpat mRNA levels) are modulated by fasting-refeeding in an Acbp6-dependent manner. There are 3 FASs in flies – which FAS does this antibody recognize? How about ACC mRNA levels? As the transcript levels of both FAS and ACC are known to respond rapidly to nutritional state, the authors could show these alongside with Lipin and Gpat.

Supplementary Figure 5: Please add quantifications of the immunostainings and also qPCRs of the relevant genes studied in the main figure. Especially important would be to include mRNA levels of CPT1, FAS, ACC, Gpat and Lipin in upon rescue of Acbp6 expression on the mutant background.

The authors have tested the involvement of small set of signaling pathways that would mimic Acbp6-like effect upon starvation-refeeding transition. These include the Stat92E, Foxo, Relish, HNF4, Nrf2, PGC1a and Myc. From their experiments the authors conclude that out of these, only Stat92e phenocopies Acbp6 and further that Stat92e is downstream of Acbp6 as loss of Acbp6 leads to blunted Stat92e activation upon refeeding. The study lacks the attempt to identify upstream regulators of Acbp6 that could be relatively easily done. Firstly, are there known TF binding domains in the Acbp6 promoter region? The authors should compare the promoter regions of Acbp3, 4, 5 and 6 to identify possible candidates specifically regulating Acbp6 (assuming that this is the only one regulated in this specific manner during starvation-refeeding in the midgut). The authors could test a few obvious suspects including sugarbabe and Mondo/Mlx, all known starvation responsive transcription factors that are also known to regulate lipid metabolism (in addition to the already tested HNF4 and foxo).

Recently, the hexosamine pathway (together with IIS) has been also shown to regulate intestinal stem cell nutrient sensing and proliferation – have the authors tested whether these pathways are interconnected?

The authors test whether the Acbp6-Stat92e-Upd3 axis is unique to nutrient adaptation by exposing the flies to *Pseudomonas entomophila* and show that loss of Stat92e does not lead to impaired cell proliferation upon bacterial infection. The conclusion (row 427) is however an overstatement, as while it may not be required for bacterial infection, the literature (and FlyBase expression data) points towards that it could be needed for viral infection (Acbp6 seems to be highly induced upon viral infection in larvae). However, in my opinion, this is not the focus of this study, and the authors only need to tone down the conclusions that Acbp6 is specifically nutrient responsive (as other stimuli were not really studied).

Minor comments:

Figure 1A is too small to read.

Figure S6C is not mentioned in the text at all.

The "(E)" is missing from the S7 Figure legend.

Row 415: Fig. 6D should probably be 6C.

Row 596: TARGET system lacks the reference.

Methods sections in general: please describe which homogenization method was used in each protocol.

Reviewer #2 (Remarks to the Author):

In this interesting paper, Li & Karpac detail the function of Acyl-CoA binding-protein 6 (Acbp6) in the fly gut, and make the interesting conclusion that its function is important for metabolic adaptations in the *Drosophila* gut. That an enzyme regulating the levels of Acetyl CoA – one of the most common substrates in energy and lipid metabolism, histone modification, and synthesis of different hormones, may regulate tissue plasticity is an exciting proposition. Specifically, the authors propose that re-feeding after fasting causes Acbp6 up-regulation and acetylation of unspecified proteins that activate Upd3/JAK/STAT signaling, leading to the ISC proliferation necessary for the nutrient-induced tissue re-growth. The paper doesn't identify these acetylation targets or provide a detailed analysis of the new "pathway" proposed. The authors generated three useful Acbp6 transgenic lines for overexpression, deletion, and reporter activity for this project. They also performed many genetic experiments, using tissue level and biochemical readouts, to generate data supporting most of their conclusions. In general the paper is a high quality study, and the data warrant the conclusions. But there are a few controls and obvious supporting experiments that are missing, as well as some overstated conclusions, as listed below. Overall, we find this to be an interesting, thought provoking manuscript that contributes in the fields of metabolic regulation, stem cell fate determination, and regulation enteric tissue function. With a few simple improvements it can be appropriate for Nature Communications.

We have the following specific comments and questions:

1. The paper's conclusions are based in part on expression data from mRNA-Seq and a transcriptional reporter (of questionable accuracy), showing that Acbp6 is gut specific and induced upon refeeding after fasting. But expression of the Acbp6 protein is not described. The paper could be improved a lot with protein expression data. This could confirm the mRNA expression data and also show the sub-cellular location of this interesting protein. Without protein expression data, the conclusions about expression must be clearly qualified.
2. How are we to know whether the Acbp6-RFP transcriptional reporter is accurate? Please comment in the paper text.
3. The term "metabolic cycles" is used repeatedly, but it is unclear what this means. Please either define it, or use the more general terms "metabolism", "metabolic reaction/pathway/network".
4. Please include data on the size of the Acbp6 mutant flies. Are they stunted?
5. The conclusion on lines 248-250 is really a conjecture, and should be stated as such. It is not fully supported by data, and cannot be a conclusion.
6. Controls need to be provided for the experiments in Fig 2F,G, showing that the mitochondrial dysfunction is not simply due to aging. Similar aging controls should be provided for the data in Fig S6A-B. Control data could be included in the supplement.

7. The conclusion on lines 285-287 is not warranted. Either the authors need to measure citrate directly, or qualify the conclusion.

8. In Figure 3H, it would be interesting to show Ac-Lys in aged conditions and Acbp6 mutants.

9. In Figure 4, it would be very interesting to see if acetate supplementation during fasting provides any benefit. It is also important to show whether Acetate supplementation stimulates ISC divisions in the ad libitum condition. Please provide such data if possible.

10. The paper does not resolve how increased Acbp6 transcriptionally up-regulates upd. Are there transcription factors other than Stat92E that are activated through acetylation, which in turn up-regulate Upd3?

11. Overall, we do not think the phenotypes observed in the Acbp6 over-expression conditions can be explained completely due to Upd3-mediated signaling. Both Acbp6 and STAT92e knockdown are associated with Upd3 transcriptional downregulation. While Upd3 is a known activator for JAK/STAT signaling, in this context it is unclear whether Upd3 is upstream or downstream of STAT92e. Is it possible that STAT92e is directly acetylated in Acbp6 gain of function conditions, and then transcriptionally up-regulates Upd3? That would explain the partial requirement for Upd3 for Acbp6-induced ISC proliferation in Figure 6G. Please comment on these possibilities in the discussion.

12. The authors should show which cell type(s) the JAK/STAT signaling is activated in using cell type-specific markers. Is it restricted to ISCs or EBs? The 10XSTAT-GFP positive cells look too big to be ISCs.

13. The authors might consider moving the NP1-Gal4ts- UAS Acbp6 +/- UAS STATRNAi data in Figure 6 E and F to Figure 5 to make a clear case for the Acbp6-Jak/Stat signaling circuit.

14. In figure 3 E, lines 298- 300, the phenotypes look like partial rescues, rather than "...completely limits the restoration" as the authors assert. A quantification would be helpful to support the conclusion, which should otherwise be limited.

15. We find the protein acetylation data to be weak in Figure S6A-B. The effects are not robust. Please comment.

16. Figure 4 E: In Acbp6RNAi + refeeding without acetate condition, are the acetylated cell numbers significantly different from the control? Please comment.

17. In Figure 4F, the acetate feeding gives only partial rescue. The number of observations (~9) seems too small to have strong confidence in the differences. Please qualify the conclusions or provide more compelling data.

18. Figure S6-C is not mentioned or discussed in the text. This figure has hardly any EsgGFP+ cells in the ad libitum condition. While the acetylation effect is robust, the absence of EsgGFP cells looks weird. Please comment, and add a description in the text.

19. The authors started by showing the differential expression of Acbp6 mRNA in response to aging, infection, microbiota, and injury. They show that the Acbp6-Acetylation- Jak/Stat signaling circuit is associated with ISC proliferation and tissue plasticity. However, they also found that Stat92E is not necessary in ECs for enteric infection-induced ISC proliferation. Does this mean that the Acbp6-Jak/Stat circuit is exclusive to nutrient-mediated ISC proliferation? In addition, aging is a chronic cue, unlike acute cues such as infection, injury, or nutrition. The discussion should include comments about whether chronic (aging) cues also activate Acbp6- Jak/Stat pathway (the Jasper lab has published data on this).

Minor comments:

1. The title should include the name of the enzyme Acbp6 to make it more informative.

2. The use of more specific words would be helpful: Examples: Lines 242 and 245 - “attenuation” could mean anything, such as mutational changes, chemical inhibition, or knockdown. Replacing “attenuation” with “RNAi-mediated knock down” or “mutation” would specifically indicate how the authors have attenuated their genes of interest and help readers interpret the specific data properly.

3. The manuscript requires a few spelling and syntax corrections:

A. Lines 236 and 237, the words “metabolic cycles” are unnecessary

B. Line 269, typo, should be “deprivation” rather than “depravation”

C. Line 281, “.., where is freely...” should be “.., where it freely...”

D. Line 342, typo, in “tunning”. The authors may consider rephrasing the sentence to make it simple.

Reviewer #3 (Remarks to the Author):

“A distinct Acyl-CoA binding protein shapes tissue plasticity during nutrient adaptation in *Drosophila*,” by authors Li and Karpac, is a detailed investigation into the role of a previously uncharacterized acyl-CoA binding protein (Acbp6) in *Drosophila*. Overall, the studies were well designed and the dataset robust. The data are impactful due to uncovering a role—for a protein about which nothing was known—in regulating the plasticity of the midgut during cycles of fasting-refeeding. While this manuscript is well-written and details strong work, there are a few concerns as listed below.

RESULTS

1. In Fig 2F-K, it appears that all mitochondrial parameters measured decline in the absence of Acbp6. Is this due to loss of mitochondrial number? It would be useful to measure mitochondrial number via qPCR for mtDNA/nuclear DNA. Do these mitochondrial protein/gene abundances normalize during acetate supplementation experiments conducted for Fig 4? If so, that could indicate that loss of nuclear histone acetylation upon knockdown of Acbp6 is impacting mitochondrial biogenesis.
2. In the results shown in Fig 3B-D, the transgenic fly strain over-expressed Acbp6 ubiquitously throughout whole body, and the metabolite measures were made in whole body extracts. These data are difficult to interpret and have little bearing on the highly specific role of Acbp6 in gut metabolism.
3. Fig 3C is misleading. Citrate was not measured, but rather, the activity of the enzyme that makes citrate. CS activity is typically used as a marker of mitochondrial abundance. This cannot be used to indicate the levels of citrate.
4. In the acetate rescue experiments of Fig 4, it would be useful to add data showing the effect of acetate supplementation on midgut length and the proportion of Delta+ cells. Where these key parameters also rescued?
5. Minor concern: In several places, the data are interpreted with statements such as “...attenuating Acbp6 in enterocytes utilizing RNAi limits midgut resizing only during refeeding, suggesting acyl-CoA metabolism plays a key role in midgut plasticity during nutrient adaptation (Lines 173-176) or “spatio-temporal calibration of acyl-CoA metabolism is critical for nutrient dependent midgut plasticity” (Lines 227-228). These statements represent an over-interpretation of the findings, since at no point is it confirmed that Acbp6 actually binds to or traffics acyl-CoAs, or that it delivers them to mitochondria for

FAO (which is also not directly quantified). Thus, it remains unconfirmed that it plays a direct role in acyl-CoA metabolism. Please edit these over-interpretations.

DISCUSSION

1. The data suggest that midgut fatty acid metabolism is critical for maintaining acetyl-CoA levels (using acetylation as the marker for this) during refeeding, when presumably the gut is full of carbohydrate. Can you comment on this continued reliance upon fatty acids in the presence of carbohydrate in the fed state, which seems unusual.

2. What is known about the other 3 Acbps that exhibit similar gut-specific expression in drosophila. Clearly none of the other 3 can compensate for the loss of Acbp6. Have Acbp3, 4, or 5 been characterized? Are they also induced upon fasting and suppressed with refeeding? Please discuss if anything is known about these other gut specific Acbp genes.

REVIEWER COMMENTS

Reviewer #1 (Remarks to the Author):

This study by Li & Karpac presents a significant work on the role of Acbp6 in starvation induced tissue plasticity and acyl-CoA metabolism. The authors show right in the beginning that Acbp6 is a crucial regulator of midgut remodeling, including proliferation and acyl-CoA sequestering upon refeeding. They also show that Stat92e (and further Upd3) is acting directly or indirectly downstream of Acbp6 and is indispensable for Acbp6 regulated tissue plasticity during starvation. The experiments are well designed and performed, and the authors have utilized the many of the strengths Drosophila model has to unravel the role of Acbp6 in tissue remodeling and nutrient utilization upon starvation. The manuscript is well-written and presents an important piece of work for this field. Along with some issues the authors should correct/improve, this study could be strengthened by a few additional experiments suggested below.

Specific comments:

Row 122: How the authors have reasoned the study in the beginning is illogical. The authors state that they have “employed a small-scale meta-analysis of accessible *D. melanogaster* transcriptomes” and that these transcriptomes include ones that show changes to external cues including nutrition. However, none of the references include “nutritional” transcriptomes, but 3 of infection, 1 of microbiota and 1 of aging. Considering, that the whole focus of the study is on a nutrition-responsive gene Acbp6, the authors should include or at least discuss the existing nutritional transcriptomes and whether Acbp6 has been already shown to be regulated by nutritional cues. The reasoning between (largely) viral transcriptomes and a study of tissue plasticity during nutrition adaptation would need more clarification.

We agree that the meta-analysis is lacking distinct databases that are more directly associated with nutrient/diet changes. We had intended to perform an even larger analysis, but found Acbp6 early on and decided to pursue its function in relation to nutrient adaptation. However, we have also added additional description of other *Drosophila* genome-wide expression analyses linked to nutrient deprivation and dietary changes that show changes in Acbp6 expression within these datasets. LINE 199-202.

Furthermore, in the Discussion, we discussed the role of mTOR-mediated nutrition signaling in viral infection to better tie in these transcriptomes within the meta-analysis. LINE 904-908.

Many immunostainings lack quantification. Please add them with all figures with immunostainings. I acknowledge that this may be difficult for some cases and in these situations adding a panel of images to supplement could be an option.

We have included quantifications for all immunostainings (e.g., FASN1, Bodipy, ATP5a, and TMRE) within Supp. Figures (SuppFig.6a, b, d, e and SuppFig.7b, d, f) for both RNAi and mutant analyses. Please see data below.

While the role of Acbp6 in regulating tissue plasticity and acetyl-CoA metabolism during starvation-refeeding is very clearly shown by the authors, a big question remains on the role of Acbp6 in prolonged starvation. To clarify this profound question, the authors could include data on the starvation sensitivity of flies upon loss / overexpression of Acbp6 in the midgut (and also Acbp6 null mutants).

We performed starvation sensitivity assay in flies upon loss of Acbp6 and overexpression of Acbp6 in the midgut enterocytes, as well *acbp6* null mutants. Depletion of Acbp6 leads to decreased survival, and over-expression of Acbp6 increases survival (SuppFig.6j-k). *acbp6* null mutants display less severe phenotypes compared to Acbp6 RNAi flies, which could be due to genetic background differences. Please see data to the right.

As the authors state on row 146 that *Acbp6* is “specifically” a key regulatory node in nutrient adaptation. To make such a conclusion, the authors should show how other *Acbp3*, 4 and 5 are expressed compared to *Acbp6*, and if they are required in the midgut for tissue plasticity during starvation-refeeding in a similar manner to *Acbp6*. Moreover, this conclusion starting on row 145 is not quite in line with the results shown by this far (only the phylogenetic comparison and expression data are shown thus far) and should be corrected accordingly.

This is an excellent suggestion that we agree improves interpretation of data in the manuscript. To this end, we conducted a thorough analysis of the expression patterns of *Acbp3*, *Acbp4*, *Acbp5*, and *Acbp6* using multiple available single-cell sequencing data of the *Drosophila* gastrointestinal tract. Our findings revealed that *Acbp4* and *Acbp6* are predominantly expressed in enterocytes, whereas *Acbp3* and *Acbp5* exhibit ubiquitous expression across various gut cells (see below).

Subsequently, we investigated the impact of other *Acbps* on intestinal stem cell (ISC) proliferation during nutrient adaptation. Interestingly, when we specifically attenuated down *Acbp3* or *Acbp5* in enterocytes ($NP1Gal4>UAS-Acbp3^{RNAi}$ or $UAS-Acbp5^{RNAi}$), we observed no significant inhibition of intestinal stem cell (ISC) proliferation during refeeding (as seen with *Acbp6*). In contrast, attenuation of *Acbp4* in enterocytes ($NP1Gal4>UAS-Acbp4^{RNAi}$) resulted in a pupal lethality (a lack of eclosion into adult stages) emphasizing its role in pupal development (dissimilar from *Acbp6*). These data are incorporated into SuppFig. 5a-c, and provide additional insight into putative specificity of *Acbp6* in the regulation of nutrient-dependent tissue plasticity, which has been qualified in the test (see LINE 453-460 and data below).

Additional data related to specificity is provided in response to another comment below (linked to upstream regulators of *Acbp6*).

Conclusion (starting on row 145) is modified as “Combined with the phylogenetic evidence, these analyses suggest that *Acbp6* may represent a key metabolic regulatory node in the evolution of nutrient adaptation.” LINE 219-221.

Row 154: It would be important to note in the beginning that *Acbp6* is also expressed at high levels in the middle

midgut. To clarify this, the authors could show the expression levels of Acbp6 +/- starvation + RF, in different parts of the gut with a horizontal diagram of the fly gut.

We obtained tiled whole gut images of Acbp6-RFP flies under *ad libitum*, fasting, and refeeding conditions to illustrate the spatial distribution of Acbp6 expression (see SuppFig.2a). The images provide a more comprehensive visualization of Acbp6 expression patterns in the *Drosophila* gastrointestinal tract. The images are a bit difficult to compare because of shrinkage during fasting, so we have added labels for anterior and posterior regions. Please see data below.

a

Row 242: The authors show that FASN protein levels (and later Lipin and Gpat mRNA levels) are modulated by fasting-refeeding in an Acbp6-dependent manner. There are 3 FASs in flies – which FAS does this antibody recognize? How about ACC mRNA levels? As the transcript levels of both FAS and ACC are known to respond rapidly to nutritional state, the authors could show these alongside with Lipin and Gpat.

The anti-body recognizes the FASN1 protein (this is clarified in the text). We also investigated the expression of ACC and FASN1 in the fly midgut during fasting-refeeding. Firstly, we found that ACC expression remained relatively stable and did not exhibit significant changes during the fasting-refeeding cycles in controls or in response to depletion Acbp6 in midgut enterocytes. On the other hand, FASN1 expression showed a notable decrease during refeeding, however, when we specifically depleted Acbp6 in enterocytes, we observed a significant increase in FASN1 expression in the fly midgut (these findings generally match what we found with FASN1 immunostaining). We have included the FASN1 transcription data in SuppFig.6c, and are providing the ACC transcription data for the reviewer (see data below).

Supplementary Figure 5: Please add quantifications of the immunostainings and also qPCRs of the relevant genes studied in the main figure. Especially important would be to include mRNA levels of CPT1, FAS, ACC, Gpat and

Lipin in upon rescue of *Acbp6* expression on the mutant background.

As described above, we have included quantifications for all immunostainings (e.g., *FASN1*, Bodipy, *ATP5a*, and *TMRE*) within Supp. Figures (SuppFig.6a,b,d,e and SuppFig.7b,d,f) for both RNAi and mutant analyses.

We also preformed transcription analyses of *CPT1*, *FASN1*, *SdhA*, *Gpat*, and *Lipin* upon rescue of *Acbp6* expression in the *acbp6* mutant background, highlighting that genetic expression of *Acbp6* can rescue transcriptional changes of these genes (during fasting and/or refeeding) induced by depletion of *Acbp6*. We have included these transcription data in SuppFig.8a-e (see below).

The authors have tested the involvement of small set of signaling pathways that would mimic *Acbp6*-like effect upon starvation-refeeding transition. These include the *Stat92E*, *Foxo*, *Relish*, *HNF4*, *Nrf2*, *PGC1a* and *Myc*. From their experiments the authors conclude that out of these, only *Stat92e* phenocopies *Acbp6* and further that *Stat92e* is downstream of *Acbp6* as loss of *Acbp6* leads to blunted *Stat92e* activation upon refeeding. The study lacks the attempt to identify upstream regulators of *Acbp6* that could be relatively easily done. Firstly, are there known TF binding domains in the *Acbp6* promoter region? The authors should compare the promoter regions of *Acbp3*, *4*, *5* and *6* to identify possible candidates specifically regulating *Acbp6* (assuming that this is the only one regulated in this specific manner during starvation-refeeding in the midgut). The authors could test a few obvious suspects including *sugarbabe* and *Mondo/Mlx*, all known starvation responsive transcription factors that are also known to

regulate lipid metabolism (in addition to the already tested HNF4 and foxo).

This is an intriguing questions. To this end, we performed a comparative analysis of the promoter/enhancer regions of the *Acbp3*, *Acbp4*, *Acbp5*, and *Acbp6* genes and identified a DNA binding motif (characterized as CTACCAA) for the transcriptional regulator Schlank located uniquely in the upstream promoter region of *Acbp6* (SuppFig. 5d). Schlank is a conserved Ceramide Synthase, which functions as both an enzyme and a nutrient- and homeodomain-dependent transcriptional regulator, often acting as a sensor for lipid levels to subsequently modulate gene expression. We found that reducing Schlank function in enterocytes strongly decreases *Acbp6*-RFP reporter activity in the posterior midgut during fasting conditions (*Acbp6*-RFP, NP1Gal4>UAS-Schlank^{RNAi}; SuppFig. 5d-e). We also found that attenuation of Schlank (NP1Gal4>UAS-Schlank^{RNAi}) impedes ISC proliferation during fasting-refeeding transitions (SuppFig. 5f). Finally, investigating the transcriptional regulatory role of Schlank on different *Acbp* paralogs revealed that only *Acbp6* expression is diminished following Schlank attenuation in enterocytes (SuppFig. 5g). Collectively, these findings indicate that the Ceramide Synthase Schlank serves as an potential upstream regulator of *Acbp6* during nutrient adaptation in the midgut. We have added these data to SuppFig.5d-g), and modified the text accordingly (LINE 460-478). See data below.

Recently, the hexosamine pathway (together with IIS) has been also shown to regulate intestinal stem cell nutrient sensing and proliferation – have the authors tested whether these pathways are interconnected?

We have not directly tested whether these pathways are integrated. But this does raise an interesting question related to the interconnectedness of metabolic networks and nutrient sensing in enterocytes and intestinal stem cells that are likely coordinated to shape tissue regeneration and plasticity. We have thus added this to the discussion. LINE 959-965.

The authors test whether the *Acbp6*-Stat92e-Upd3 axis is unique to nutrient adaptation by exposing the flies to *Pseudomonas entomophila* and show that loss of Stat92e does not lead to impaired cell proliferation upon bacterial infection. The conclusion (row 427) is however an overstatement, as while it may not be required for bacterial infection, the literature (and FlyBase expression data) points towards that it could be needed for viral infection

(Acbp6 seems to be highly induced upon viral infection in larvae). However, in my opinion, this is not the focus of this study, and the authors only need to tone down the conclusions that Acbp6 is specifically nutrient responsive (as other stimuli were not really studied).

We absolutely agree, and have thus revised the conclusion/interpretation of these data and modified the text accordingly. LINE 860-865.

Minor comments:

Figure 1A is too small to read.

We enlarged the font size in Fig.1A as much as possible.

Figure S6C is not mentioned in the text at all.

Fig. S6C was mentioned in line 298 (original version of manuscript) as “Pan-lysine acetylation is revealed throughout midgut cell types, but is most prominent in enterocyte nuclei (minimally in enterocyte cytoplasm, Fig. 3E and S6C).” This is now SuppFig.9a)

The “(E)” is missing from the S7 Figure legend.

This has been added to the new Supplemental Figure.

Row 415: Fig. 6D should probably be 6C.

This has been corrected.

Row 596: TARGET system lacks the reference.

A reference was added for the genetic TARGET system.

Methods sections in general: please describe which homogenization method was used in each protocol.

We have added more description regarding our homogenization methods.

Reviewer #2 (Remarks to the Author):

In this interesting paper, Li & Karpac detail the function of Acyl-CoA binding-protein 6 (Acbp6) in the fly gut, and make the interesting conclusion that its function is important for metabolic adaptations in the Drosophila gut. That an enzyme regulating the levels of Acetyl CoA – one of the most common substrates in energy and lipid metabolism, histone modification, and synthesis of different hormones, may regulate tissue plasticity is an exciting proposition. Specifically, the authors propose that re-feeding after fasting causes Acbp6 up-regulation and acetylation of unspecified proteins that activate Upd3/JAK/STAT signaling, leading to the ISC proliferation necessary for the nutrient-induced tissue re-growth. The paper doesn't identify these acetylation targets or provide a detailed analysis of the new “pathway” proposed. The authors generated three useful Acbp6 transgenic lines for overexpression, deletion, and reporter activity for this project. They also performed many genetic experiments, using tissue level and biochemical readouts, to generate data supporting most of their conclusions. In general the paper is a high quality study, and the data warrant the conclusions. But there are a few controls and obvious supporting experiments that are missing, as well as some overstated conclusions, as listed below. Overall, we find this to be an interesting, thought provoking manuscript that contributes in the fields of metabolic regulation, stem cell fate determination, and regulation enteric tissue function. With a few simple improvements it can be appropriate for Nature Communications.

We have the following specific comments and questions:

1. The paper's conclusions are based in part on expression data from mRNA-Seq and a transcriptional reporter (of questionable accuracy), showing that Acbp6 is gut specific and induced upon refeeding after fasting. But expression of the Acbp6 protein is not described. The paper could be improved a lot with protein expression data. This could confirm the mRNA expression data and also show the sub-cellular location of this interesting protein. Without protein expression data, the conclusions about expression must be clearly qualified.

We agree that the study focuses on transcription of Acbp6. As of yet, we do not have a reliable molecular tool to investigate endogenous Acbp6 protein levels. To the end, we have:

- (I) Made sure to clarify and qualify through-out the text that our analyses are only related to gene expression/transcription.
- (II) Identified an upstream regulator of *Acbp6* to help clarify nutrient-dependent transcriptional changes (see Comment 2 below).

2. How are we to know whether the *Acbp6*-RFP transcriptional reporter is accurate? Please comment in the paper text.

To address this question, we attempted to identify potential upstream regulators of *Acbp6* that could be target to change reporter activity and reflect regulatability (although it must be noted that these types of reporters only provide a window into regulation). We performed a comparative analysis of the promoter/enhancer regions of the *Acbp3*, *Acbp4*, *Acbp5*, and *Acbp6* genes and identified a DNA binding motif (characterized as CTACCAA) for the transcriptional regulator *Schlank* located uniquely in the upstream promoter region of *Acbp6* (SuppFig. 5d). *Schlank* is a conserved Ceramide Synthase, which functions as both an enzyme and a nutrient- and homeodomain-dependent transcriptional regulator, often acting as a sensor for lipid levels to subsequently modulate gene expression. We found that reducing *Schlank* function in enterocytes strongly decreases *Acbp6*-RFP reporter activity in the posterior midgut during fasting conditions (*Acbp6*-RFP, NP1Gal4>UAS-*Schlank*^{RNAi}; SuppFig. 5d-e). We also found that attenuation of *Schlank* (NP1Gal4>UAS-*Schlank*^{RNAi}) impedes ISC proliferation during fasting-refeeding transitions (SuppFig. 5f). Finally, investigating the transcriptional regulatory role of *Schlank* on different *Acbp* paralogs revealed that only *Acbp6* expression is diminished following *Schlank* attenuation in enterocytes (SuppFig. 5g). Collectively, these findings indicate that the Ceramide Synthase *Schlank* serves as a potential upstream regulator of *Acbp6* during nutrient adaptation in the midgut, and give further evidence of reporter accuracy. We have added these data to SuppFig.5d-g, and modified the text accordingly (LINE 460-478). See data below.

3. The term “metabolic cycles” is used repeatedly, but it is unclear what this means. Please either define it, or use the more general terms “metabolism”, “metabolic reaction/pathway/network”.

We agree, and have made changes through-out the text, replacing “metabolic cycles” with “metabolic pathways” or “metabolic network/s”.

4. Please include data on the size of the *Acbp6* mutant flies. Are they stunted?

Through all gross anatomical observations, *acbp6* mutant flies and *WT* controls look similar (no change in size). We have included these in SuppFig.4d. See data below.

5. The conclusion on lines 248-250 is really a conjecture, and should be stated as such. It is not fully supported by data, and cannot be a conclusion.

We agree and have qualified these statements to better reflect the data without additional conjecture. LINE 516-518.

6. Controls need to be provided for the experiments in Fig 2F,G, showing that the mitochondrial dysfunction is not simply due to aging. Similar aging controls should be provided for the data in Fig S6A-B. Control data could be included in the supplement.

It is important to note that all experiments are performed rigorously on young flies. In other words, after mating and *ad libitum* feeding, 2 days fasting, and 2 days refeeding, these animals are all still 8-10 days old. However, we have added additional controls showing that flies fed *ad libitum* for an additional 4 days (to match the 2 days of fasting and 2 days of refeeding) do not display mitochondrial dysfunction (ATP5A and TMRE immunostains in controls and upon depletion of *Acbp6* [RNAi] in midgut enterocytes). We have included these in SuppFig.6f-g. See data below.

7. The conclusion on lines 285-287 is not warranted. Either the authors need to measure citrate directly, or qualify the conclusion.

We completely agree, and have modified the text, Figure, and Figure Legend to reflect we are measuring Citrate Synthase Activity, and not Citrate directly.

8. In Figure 3H, it would be interesting to show Ac-Lys in aged conditions and Acbp6 mutants.

It is intriguing to consider the role of Acbp6 in tissue aging. However, aging is not the focus of the manuscript, and would likely need an independent analysis to draw firm conclusions about Acbp6, acetyl-CoA metabolism, and tissue aging. To this end, we performed Ac-Lys midgut immunostains with aged flies (50 days old) and found that aging alone can increase the intensity of Ac-Lys staining (likely part of tissue dysplasia), regardless of whether the flies are under *ad libitum* or fasting conditions (see below). Furthermore, we observed that flies with attenuated Acbp6 expression in enterocytes (NP1Gal4>UAS-Acbp6^{RNAi}) also exhibited relatively high levels of Ac-Lys intensity during aging (see below). These findings strongly suggest that the aging process itself drives changes in pan-acetylation levels (and potentially acetyl-CoA metabolism) in the midgut, which could be independent of Acbp6 function. We have included these data for the reviewer (see below).

9. In Figure 4, it would be very interesting to see if acetate supplementation during fasting provides any benefit. It is also important to show whether Acetate supplementation stimulates ISC divisions in the *ad libitum* condition. Please provide such data if possible.

To address this, we performed Ac-Lys midgut immunostainings, as well as pH3+ cell quantification, during fasting with/without acetate supplementation in both control flies and flies with enterocyte-specific Acbp6 depletion. Our results demonstrated that acetate supplementation during fasting had minimal effects on pan-acetylation levels, or intestinal stem cell (ISC) proliferation. We have included these in SuppFig.9e-f. See data below.

Additionally, pH3+ cell quantification was performed under the *ad libitum* conditions with acetate supplementation, which showed that acetate supplementation alone cannot induce ISC proliferation (see data below merged into Figure 4f).

10. The paper does not resolve how increased Acbp6 transcriptionally up-regulates upd. Are there transcription factors other than Stat92E that are activated through acetylation, which in turn up-regulate Upd3?

This is an intriguing question. There are likely to be numerous other signaling pathways/transcription factors tuned by acetylation (we have qualified this in text as well). So we cannot rule out other pathways involved in Acbp6-STAT92E control of Upd3 transcription in enterocytes. In previous studies, it has been extensively demonstrated that enterocyte-derived Upd cytokines alone are sufficient to stimulate ISC proliferation in the Drosophila midgut. Therefore, considering our finding that Acbp6 expression in enterocytes can regulate ISC proliferation, we hypothesized that Acbp6 might exert its effect by targeting Upds. To investigate this further, we examined the regulatory role of other transcription factors, including Relish and Egfr, on the expression of Upd3. However, when we specifically attenuated Relish or Egfr in midgut enterocytes (NP1Gal4>UAS-Relish^{RNAi} or Egfr^{RNAi}), we did not observe a decrease in Upd3 expression during refeeding (which we observe with depletion of Acbp6 and STAT92E in enterocytes, Fig. 6). These findings, which only serve as negative controls but highlight the ability of both Acbp6 and STAT92E to regulate Upd3, have been incorporated into SuppFig.10c to provide a more comprehensive view of our findings. See data below.

11. Overall, we do not think the phenotypes observed in the Acbp6 over-expression conditions can be explained completely due to Upd3-mediated signaling. Both Acbp6 and STAT92e knockdown are associated with Upd3 transcriptional downregulation. While Upd3 is a known activator for JAK/STAT signaling, in this context it is unclear whether Upd3 is upstream or downstream of STAT92e. Is it possible that STAT92e is directly acetylated in Acbp6 gain of function conditions, and then transcriptionally up-regulates Upd3? That would explain the partial requirement for Upd3 for Acbp6-induced ISC proliferation in Figure 6G. Please comment on these possibilities in the discussion.

Numerous studies have provided evidence demonstrating that enterocyte-derived Upd3 plays a crucial role in

activating JAK/STAT signaling in intestinal stem cells (ISCs). However, the regulatory mechanism we uncovered likely differs from the canonical model, where JAK/STAT signaling activity in enterocytes regulates (upstream) Upd3 expression, potentially even directly as a transcriptional activator. As pointed out, we agree that there may be a direct link between Acbp6 Gain-of-Function and the acetylation of STAT92e, leading to the transcriptional up-regulation of Upd3 directly by Upd3. Since we don't have a molecular tool to measure STAT acetylation in flies, this can only be a hypothesis (although a strong one). To further clarify this possibility, we have included additional description in the Discussion, stating that "This suggests that STAT92e may be directly acetylated by Acbp6-dependent acetyl-CoA abundance and sequential acetylation to transcriptionally regulating Upd3 expression and secretion" (LINE 1027-1029). This revised statement helps clarify a potential mechanism by which Acbp6 influences Upd3 expression and JAK/STAT signaling activation.

12. The authors should show which cell type(s) the JAK/STAT signaling is activated in using cell type-specific markers. Is it restricted to ISCs or EBs? The 10XSTAT-GFP positive cells look too big to be ISCs.

We uncovered that JAK/STAT signaling is activated in enterocytes (large polyploid cells) after refeeding, and is maintained in intestinal stem cells (ISCs) as well. We employed a Delta antibody as a marker for ISCs and performed immunostaining analysis. Interestingly, we observed that GFP signals, indicative of STAT-GFP reporter activity, were not confined solely to Delta-positive cells but were also present in enterocytes, characterized by larger, nuclei following refeeding. As we mentioned previously, "STAT-GFP reporter activity is maintained in stem/progenitor cells but increased in large, polyploid enterocytes during refeeding (2 days, Fig. 5c and S7a [now 10b])." However, when we attenuated Acbp6 specifically in enterocytes (NP1Gal4>UAS-Acbp6^{RNAi}), we observed a restriction of this nutrient-dependent STAT activation, as evidenced by a decrease in STAT-GFP intensity in large, polyploid enterocytes. These findings highlight the crucial role of Acbp6 in mediating nutrient-dependent activation of the JAK/STAT signaling pathway, leading to its dynamic regulation in enterocytes during the refeeding process. These data also highlight the potential unique regulation of Upd3 (via STAT in enterocytes) described in the Comment above. See data below.

13. The authors might consider moving the NP1-Gal4ts- UAS Acbp6 +/- UAS STATRNAi data in Figure 6 E and F to Figure 5 to make a clear case for the Acbp6-Jak/Stat signaling circuit.

We agree this could aid in transition of ideas, but we found that it made Figure 5 a bit overwhelming in size and required showing control data multiple times, so we decided the best option was to use the current format.

14. In figure 3 E, lines 298- 300, the phenotyped look like partial rescues, rather than "...completely limits the restoration ..." as the authors assert. A quantification would be helpful to support the conclusion, which should otherwise be limited.

We apologize for the confusion. The quantification is present in Fig. 4E in relation to +/- acetate supplementation. From the histogram, the lack of restoration of Acyl-Lysine when *Acbp6* is depleted in midgut enterocytes is clear. We have updated the text, and changed "(Fig. 3E)" to "(Fig. 3e, 4e)".

15. We find the protein acetylation data to be weak in Figure S6A-B. The effects are not robust. Please comment.

Given the number of different genetic backgrounds and conditions we have explored pan acetylation immunostaining, all finding similar significant trends, we believe these data are reasonable (and significant), even though not as robust as other experiments. This could be due to genetic background effects that lead to some variation in nutrient adaptation.

16. Figure 4 E: In *Acbp6*RNAi + refeeding without acetate condition, are the acetylated cell numbers significantly different from the control? Please comment.

We apologize for the confusion. Yes, the p value is less than 0.001, and we have added a "***" label in the Figure.

17. In Figure 4F, the acetate feeding gives only partial rescue. The number of observations (~9) seems too small to have strong confidence in the differences. Please qualify the conclusions or provide more compelling data.

We agree, and have stated that "acetate supplementation can also re-establish, in part, ISC activation/proliferation during refeeding (2 days) which is limited when *Acbp6* is attenuated in midgut enterocytes (Fig. 4F)." in the text.

18. Figure S6-C is not mentioned or discussed in the text. This figure has hardly any EsgGFP+ cells in the *ad libitum* condition. While the acetylation effect is robust, the absence of EsgGFP cells looks weird. Please comment, and add a description in the text.

Fig. S6C (now SuppFig.9a) was mentioned originally as "Pan-lysine acetylation is revealed throughout midgut cell types, but is most prominent in enterocyte nuclei (minimally in enterocyte cytoplasm, Fig. 3E and S6C)." in the original manuscript. This same description applies to the revised version.

We agree the GFP signal is weak in the *ad libitum* samples (although it is present). Admittedly, some of the GFP signal is 'drowned out' by DAPI and Pan acyl-Lysine. However, in homeostatic conditions, it is normal to see very few EsgGFP+ cells (ISCs and EBs) in a healthy midgut. This is further confirmed by our quantification of Delta positive cells in *ad libitum* fed (young) animals, which are very low. The data still show (especially in refeeding conditions when stem cell numbers are increased) that acyl-Lysine is present in non-GFP, large nuclei enterocytes.

19. The authors started by showing the differential expression of *Acbp6* mRNA in response to aging, infection, microbiota, and injury. They show that the *Acbp6*-Acetylation- Jak/Stat signaling circuit is associated with ISC proliferation and tissue plasticity. However, they also found that Stat92E is not necessary in ECs for enteric infection-induced ISC proliferation. Does this mean that the *Acbp6*-Jak/Stat circuit is exclusive to nutrient-mediated ISC proliferation? In addition, aging is a chronic cue, unlike acute cues such as infection, injury, or nutrition. The discussion should include comments about whether chronic (aging) cues also activate *Acbp6*- Jak/Stat pathway (the Jasper lab has published data on this).

Based on our data, we believe that an *Acbp6*-JAK/STAT circuit may be crucial to nutrient-mediated intestinal stem cell proliferation and tissue remodeling (different from immune responses to enteric infection, as an example). We cannot exclude that other stress-induced responses (e.g., viral infection or aging) are not also mediated by an *Acbp6*-JAK/STAT circuit, whether this involves nutrient sensing or not. To this end, we have incorporated additional discussion in the text to include other hypotheses, especially linked to aging in the *Drosophila* midgut, where JAK/STAT plays an important role in controlling dysplasia through multiple mechanisms. LINE 1033-1039.

Minor comments:

1. The title should include the name of the enzyme *Acbp6* to make it more informative.

We have added more data to the manuscript in order to highlight the specificity of *Acbp6* (distinct from other *Acbp* paralogs) in nutrient-dependent tissue plasticity. We conducted a thorough analysis of the expression patterns of *Acbp3*, *Acbp4*, *Acbp5*, and *Acbp6* using multiple available single-cell sequencing data of the *Drosophila* gastrointestinal tract. Our findings revealed that *Acbp4* and *Acbp6* are predominantly expressed in enterocytes, whereas *Acbp3* and *Acbp5* exhibit ubiquitous expression across various gut cells (see below).

Subsequently, we investigated the impact of other *Acbps* on intestinal stem cell (ISC) proliferation during nutrient adaptation. Interestingly, when we specifically attenuated down *Acbp3* or *Acbp5* in enterocytes (NP1Gal4>UAS-*Acbp3*^{RNAi} or UAS-*Acbp5*^{RNAi}), we observed no significant inhibition of intestinal stem cell (ISC) proliferation during refeeding (as seen with *Acbp6*). In contrast, attenuation of *Acbp4* in enterocytes (NP1Gal4>UAS-*Acbp4*^{RNAi}) resulted in a pupal lethality (a lack of eclosion into adult stages) emphasizing its role in pupal development (dissimilar from *Acbp6*). These data are incorporated into SuppFig.6a-c, and provide additional insight into putative specificity of *Acbp6* in the regulation of nutrient-dependent tissue plasticity, which has been qualified in the test (see LINE 453-460 and data below).

We believe these new data, coupled with other new data highlighting specificity in upstream regulation of Acbp6 compared to other Acbp paralogs, justifies the title.

2. The use of more specific words would be helpful: Examples: Lines 242 and 245 - “attenuation” could mean anything, such as mutational changes, chemical inhibition, or knockdown. Replacing “attenuation” with “RNAi-mediated knock down” or “mutation” would specifically indicate how the authors have attenuated their genes of interest and help readers interpret the specific data properly.

We have attempted to clarify this in the manuscript. Explicitly stating differences between RNAi attenuation/depletion/inhibition (by always putting an RNAi genotype after the description) and mutational analysis (always stating when we are utilizing a loss-of-function mutant).

3. The manuscript requires a few spelling and syntax corrections:

A. Lines 236 and 237, the words “metabolic cycles” are unnecessary

B. Line 269, typo, should be “deprivation” rather than “depravation”

C. Line 281, “.., where is freely...” should be “.., where it freely...”

D. Line 342, typo, in “tunning”. The authors may consider rephrasing the sentence to make it simple.

We have made these corrections.

Reviewer #3 (Remarks to the Author):

“A distinct Acyl-CoA binding protein shapes tissue plasticity during nutrient adaptation in *Drosophila*,” by authors Li and Karpac, is a detailed investigation into the role of a previously uncharacterized acyl-CoA binding protein (Acbp6) in *Drosophila*. Overall, the studies were well designed and the dataset robust. The data are impactful due to uncovering a role--for a protein about which nothing was known—in regulating the plasticity of the midgut during cycles of fasting-refeeding. While this manuscript is well-written and details strong work, there are a few concerns as listed below.

RESULTS

1. In Fig 2F-K, it appears that all mitochondrial parameters measured decline in the absence of Acbp6. Is this due to loss of mitochondrial number? It would be useful to measure mitochondrial number via qPCR for mtDNA/nuclear DNA.

We couldn't accurately quantitate mtDNA amount in dissected midguts due to the heterogeneity of the tissue and due to Acbp6 only being required to control these metabolic networks in the posterior midgut (i.e. we had to rely on cell-specific imaging). Alternatively, we performed additional experiments utilizing UAS-mito-GFP. We leveraged mito-GFP (a human mitochondrial import sequence [hCox8A] fused to GFP) as a marker of mitochondrial number and density, and found in controls that mitochondrial numbers decrease during fasting and increase upon refeeding, reflecting the dynamics of mitochondrial adaptation to nutrient availability in the midgut. Furthermore, inhibiting Acbp6 function in enterocytes (UAS-mito-GFP, NP1Gal4>UAS-Acbp6^{RNAi}) impedes the recovery of mitochondrial numbers after refeeding, suggesting a critical role for Acbp6 in regulating mitochondrial number/density in response to nutrient fluctuations. These data were incorporated into SuppFig.6h-i and are shown below.

Do these mitochondrial protein/gene abundances normalize during acetate supplementation experiments conducted for Fig 4? If so, that could indicate that loss of nuclear histone acetylation upon knockdown of *Acbp6* is impacting mitochondrial biogenesis.

We had originally performed these experiments, and found that acetate supplementation does not significantly restore mitochondrial numbers/adaptation/density upon *Acbp6* depletion in enterocytes. It's unclear if this is a more universal phenotype, or specific to *Acbp6*. It is very interesting, and helps establish the roles of different metabolite inputs into these cycles, but we hesitate to over-interpret this negative result. We are providing the data for the reviewer (see below, an ATP5A immunostain is shown).

2. In the results shown in Fig 3B-D, the transgenic fly strain over-expressed *Acbp6* ubiquitously throughout whole body, and the metabolite measures were made in whole body extracts. These data are difficult to interpret and have little bearing on the highly specific role of *Acbp6* in gut metabolism.

We agree that whole body measurements are not ideal, but in the absence of being able to distinctly measure acyl-CoA metabolism/flux in the midgut *in vivo* (as discussed below) we believe these data provide at least a little evidence of *Acbp6* -mediated changes in metabolites or enzymatic activity. Measuring metabolite levels in the *Drosophila* intestine is a challenge considering their low abundance and functional heterogeneity of cells in the tissue. While we acknowledge that the use of the ubiquitously expressed *DaGal4* driver may not be specific enough, we attempted to use the *NP1Gal4* driver to repress *Acbp6* expression specifically in enterocytes in order to observe the resulting changes. Unfortunately, due to the low levels of metabolites in dissected midguts, we were unable to detect significant differences using this driver. As a result, we resorted to using the *DaGal4* driver to overexpress *Acbp6* and assay increases in key metabolites. We recognize the limitations of this approach and have tried to address this in the text.

3. Fig 3C is misleading. Citrate was not measured, but rather, the activity of the enzyme that makes citrate. CS activity is typically used as a marker of mitochondrial abundance. This cannot be used to indicate the levels of citrate.

We completely agree, and have modified the text, Figure, and Figure Legend to reflect we are measuring Citrate Synthase Activity, and not Citrate directly.

4. In the acetate rescue experiments of Fig 4, it would be useful to add data showing the effect of acetate supplementation on midgut length and the proportion of Delta+ cells. Where these key parameters also rescued?

To address this concern, we measured midgut length and performed Delta immunostains after acetate supplementation, and found acetate supplementation can increase midgut length and elevate intestinal stem cell numbers (Delta + cells) after refeeding in flies where *Acbp6* is depleted in midgut enterocytes (NP1Gal4>UAS-*Acbp6*^{RNAi}). These are partial rescues (similar to pH3+ cells already provided). These data were incorporated into Fig. 4g and SuppFig.9g, and are shown below.

5. Minor concern: In several places, the data are interpreted with statements such as "...attenuating *Acbp6* in enterocytes utilizing RNAi limits midgut resizing only during refeeding, suggesting acyl-CoA metabolism plays a key role in midgut plasticity during nutrient adaptation (Lines 173-176) or "spatio-temporal calibration of acyl-CoA metabolism is critical for nutrient dependent midgut plasticity" (Lines 227-228). These statements represent an over-interpretation of the findings, since at no point is it confirmed that *Acbp6* actually binds to or traffics acyl-CoAs, or that it delivers them to mitochondria for FAO (which is also not directly quantified). Thus, it remains unconfirmed that it plays a direct role in acyl-CoA metabolism. Please edit these over-interpretations.

We completely agree, and had attempted in the original version of the manuscript to limit over-interpretation of acyl-CoA metabolism (which we cannot directly verify *in vivo* in the midgut), and instead focus on *Acbps* and acetyl-CoA metabolism. However, as the reviewer points out, there are still instances through-out the text where there is too much focus on acyl-CoA metabolism. We have re-evaluated the text and adjusted the conclusions accordingly to reflect a more cautious interpretation.

DISCUSSION

1. The data suggest that midgut fatty acid metabolism is critical for maintaining acetyl-CoA levels (using acetylation as the marker for this) during refeeding, when presumably the gut is full of carbohydrate. Can you comment on this continued reliance upon fatty acids in the presence of carbohydrate in the fed state, which seems unusual.

This is a very intriguing question. Other studies have shed light on the significance of fatty acids as a crucial source of acetyl-CoA, which plays a vital role in acetylation processes. In a Cell Reports¹ paper from 2016, researchers employed ¹³C-carbon tracing and acetyl-proteomics techniques to demonstrate that carbons from fatty acids can contribute up to 90% of acetylation on specific histone lysines, even when excess glucose is present. Moreover, another study published in JBC² revealed that *de novo* fatty acid synthesis and acetylation utilize the same reservoir of acetyl-CoA, as inhibiting fatty acid synthesis resulted in hyperacetylation of histones. These findings provide compelling evidence for the interconnection between fatty acid metabolism and acetylation, even in the presence of carbohydrates.

Reference:

1. McDonnell E., et al. Lipids reprogram metabolism to become a major carbon source for histone acetylation. Cell reports, 2016, 17(6): 1463-1472.
2. Galdieri L, Vancura A. Acetyl-CoA carboxylase regulates global histone acetylation. Journal of Biological Chemistry, 2012, 287(28): 23865-23876.

2. What is known about the other 3 Acbps that exhibit similar gut-specific expression in drosophila. Clearly none of the other 3 can compensate for the loss of Acbp6. Have Acbp3, 4, or 5 been characterized? Are they also induced upon fasting and suppressed with refeeding? Please discuss if anything is known about these other gut specific Acbp genes.

We have addressed this, in part, in the revised manuscript. We conducted a thorough analysis of the expression patterns of Acbp3, Acbp4, Acbp5, and Acbp6 using multiple available single-cell sequencing data of the Drosophila gastrointestinal tract. Our findings revealed that Acbp4 and Acbp6 are predominantly expressed in enterocytes, whereas Acbp3 and Acbp5 exhibit ubiquitous expression across various gut cells (see below).

Subsequently, we investigated the impact of other Acbps on intestinal stem cell (ISC) proliferation during nutrient adaptation. Interestingly, when we specifically attenuated down Acbp3 or Acbp5 in enterocytes (NP1Gal4>UAS-Acbp3^{RNAi} or UAS-Acbp5^{RNAi}), we observed no significant inhibition of intestinal stem cell (ISC) proliferation during refeeding (as seen with Acbp6). In contrast, attenuation of Acbp4 in enterocytes (NP1Gal4>UAS-Acbp4^{RNAi}) resulted in a pupal lethality (a lack of eclosion into adult stages) emphasizing its role in pupal development (dissimilar from Acbp6). These data are incorporated into SuppFig.6a-c, and provide additional insight into putative specificity of Acbp6 in the regulation of nutrient-dependent tissue plasticity, which has been qualified in the test

(see LINE 453-460 and data below).

We also performed a comparative analysis of the promoter/enhancer regions of the *Acbp3*, *Acbp4*, *Acbp5*, and *Acbp6* genes and identified a DNA binding motif (characterized as CTACCAA) for the transcriptional regulator Schlank located uniquely in the upstream promoter region of *Acbp6* (SuppFig. 5d). Schlank is a conserved Ceramide Synthase, which functions as both an enzyme and a nutrient- and homeodomain-dependent transcriptional regulator, often acting as a sensor for lipid levels to subsequently modulate gene expression. We found that reducing Schlank function in enterocytes strongly decreases *Acbp6*-RFP reporter activity in the posterior midgut during fasting conditions (*Acbp6*-RFP, NP1Gal4>UAS-Schlank^{RNAi}; SuppFig. 5d-e). We also found that attenuation of Schlank (NP1Gal4>UAS-Schlank^{RNAi}) impedes ISC proliferation during fasting-refeeding transitions (SuppFig. 5f). Finally, investigating the transcriptional regulatory role of Schlank on different *Acbp* paralogs revealed that only *Acbp6* expression is diminished following Schlank attenuation in enterocytes (SuppFig. 5g). Collectively, these findings indicate that the Ceramide Synthase Schlank serves as an potential upstream regulator of *Acbp6* during nutrient adaptation in the midgut. We have added these data to SuppFig.5d-g, and modified the text accordingly (LINE 460-478). See data below.

REVIEWER COMMENTS

Reviewer #1 (Remarks to the Author):

The authors have addressed all of my comments successfully but are asked to correct the following minor issues and add information on few details:

Line 196: Please remove the word “nutrition” as the transcriptomes you have included in the meta-analysis in Supplementary Fig. 1a do not include any nutritional ones.

Line 330: “knocking down” change to “knockdown”

Line 477: “Ceramide Synthase” change to lowercase

Line 580: word “fasting” in the Figure reference is not needed. Reconsider changing the wording of the previous sentence from “nutrient deprivation” to “starvation” as this is the experiment performed.

Please also add details of the comparative analysis of the promoter/enhancer regions of Acbp3, Acbp4, Acbp5, and Acbp6 to Materials and Methods.

In Supplementary Fig.4d it is shown that there is no size difference between acbp6 mutant flies and WT controls with a representative figure where, in my opinion, the mutants in fact look bigger. Could you provide a quantification to support the conclusion?

Reviewer #2 (Remarks to the Author):

This revised paper from Li & Karpac details (as before) the function of a putative Acyl-CoA binding-protein (Acbp6) in the fly gut, and shows that this gene is important for adaptation of the gut to starvation and refeeding. The authors propose Acbp6 is upregulated by starvation, that this affects the levels of Acetyl CoA which regulate, in turn, tissue adaptation. This is a potentially novel and interesting metabolic regulatory mechanism, and the data in the paper are consistent with it. The revised manuscript has addressed most of the reviewers’ criticisms with editorial changes and new data, and the authors responses to the reviews are in large part satisfactory. Some of the new data add to the story, for instance the effects on mitochondria shown in new Fig S6hi. However, there are still three problems with the presentation that I feel are important to address because they bear directly on the credibility of the paper’s central thesis and message:

1. First, although the authors' proposal is that starvation/re-feeding-dependent changes in Acbp6 activity are essential for the gut to recover from starvation, the data showing such changes does not meet standards in the field and is, in this reviewer's opinion, insufficient to support the author's central thesis. Although the authors propose that upregulation of Acbp6 levels in the gut is starvation dependent, the data supporting this use only an Acbp6-RFP reporter gene (Fig 1c, Fig S2a-e), which has not been validated to accurately read out Acbp6 mRNA expression. Moreover, these data were not quantitated, and the data in Fig S2a-e don't clearly show the starvation induction the authors discuss. The published RNAseq data suggested to be relevant (Fig S1, refs 44-48) is not actually relevant to starvation/refeeding in the gut. To make their point and validate their reporter, the authors really need to at least present some qRT-PCR or in situ hybridization data that documents Acbp6 mRNA expression in the gut during feeding/fasting transitions. Moreover, reporter gene and mRNA data don't necessarily reflect true levels of protein expression or activity. Ideally, the authors would measure Acbp6 protein expression using an antibody (which they could make), and also assay its activity somehow, perhaps by measurements of bound Acyl-CoA. Without such additional data the paper grossly overstates its conclusions about the regulation of metabolism by Acbp6. As it stands, the data show that the gene is important for adaptive plasticity, but do not show that changes in Acbp6 levels or activity occur, or that such changes regulate changes in metabolism needed for adaptive plasticity.

2. Another point relates to the acetyl-lysine assays shown in Fig 4 and Fig S9C. These are the only link in the paper to Acyl-CoA metabolism apart from the Acbp6 sequence homology. From their data the authors argue that levels of total protein acetylation decrease upon fasting. However, the data presented are either pictures, or measures of numbers of "Acetylated Lysine +" cells, or "Acetylated Lysine area". These metrics cannot be accurate reflections of the amount of protein acetylation per cell, or per unit chromatin, or per unit protein. Although the pictures look good, I would ask the authors to re-do their quantifications, measuring something that is a more accurate metric for how much total acetylation is present relative to a standard such as total protein, total DNA, or total cell volume or cell area. For instance, integrated GFP intensity/area would do. Given the way the samples were assayed, I didn't find Fig 4b, c, e or S9c convincing. It should be straightforward for the authors to re-do their quantifications using a different approach, perhaps even using the same images. This could better support their proposal that fasting and Acbp6 affect an output of Acyl-CoA metabolism.

3. I still suggest changing the Title. Neither this paper, nor any other, has actually shown that *Drosophila* Acbp6 binds Acyl-CoA, yet the title asserts this. In fact, Acbp6 is just ~50% conserved with its orthologs, ACBD7 and DBI, so there is a possibility it has some other molecular function.

Reviewer #3 (Remarks to the Author):

The authors have been very responsive to the original critiques, and the manuscript is now much stronger. I have no additional remarks.

REVIEWER COMMENTS

Reviewer #1 (Remarks to the Author):

The authors have addressed all of my comments successfully but are asked to correct the following minor issues and add information on few details:

Line 196: Please remove the word “nutrition” as the transcriptomes you have included in the meta-analysis in Supplementary Fig. 1a do not include any nutritional ones.

This has been removed.

Line 330: “knocking down” change to “knockdown”

This has been corrected.

Line 477: “Ceramide Synthase” change to lowercase

This has been corrected.

Line 580: word “fasting” in the Figure reference is not needed. Reconsider changing the wording of the previous sentence from “nutrient deprivation” to “starvation” as this is the experiment performed.

We have made these changes as suggested.

Please also add details of the comparative analysis of the promoter/enhancer regions of *Acbp3*, *Acbp4*, *Acbp5*, and *Acbp6* to Materials and Methods.

We have added details (an independent section in the Materials and Methods) regarding the comparative analysis (also highlighting software usage) of promoter/enhancer regions within the *Acbp* genes.

In Supplementary Fig.4d it is shown that there is no size difference between *acbp6* mutant flies and WT controls with a representative figure where, in my opinion, the mutants in fact look bigger. Could you provide a quantification to support the conclusion?

We have added quantitated wet weight data (see **Supplemental Fig. 4e) comparing *acbp6* mutant and WT flies, which further highlight a similarity in body size between the two genotypes. These data also support our finding that there are no noticeable gross anatomical differences between *acbp6* mutant and WT flies.**

Reviewer #2 (Remarks to the Author):

This revised paper from Li & Karpac details (as before) the function of a putative Acyl-CoA binding-protein (*Acbp6*) in the fly gut, and shows that this gene is important for adaptation of the gut to starvation and refeeding. The authors propose *Acbp6* is upregulated by starvation, that this affects the levels of Acetyl CoA which regulate, in turn, tissue adaptation. This is a potentially novel and interesting metabolic regulatory mechanism, and the data in the paper are consistent with it. The revised manuscript has addressed most of the reviewers' criticisms with editorial changes and new data, and the authors responses to the reviews are in large part satisfactory. Some of the new data add to the story, for instance the effects on mitochondria shown in new Fig S6hi. However, there are still three problems with the presentation that I feel are important to address because they bear directly on the credibility of the paper's central thesis and message:

1. First, although the authors' proposal is that starvation/re-feeding-dependent changes in *Acbp6* activity are essential for the gut to recover from starvation, the data showing such changes does not meet standards in the field and is, in this reviewer's

opinion, insufficient to support the author's central thesis. Although the authors propose that upregulation of *Acbp6* levels in the gut is starvation dependent, the data supporting this use only an *Acbp6*-RFP reporter gene (Fig 1c, Fig S2a-e), which has not been validated to accurately read out *Acbp6* mRNA expression. Moreover, these data were not quantitated, and the data in Fig S2a-e don't clearly show the starvation induction the authors discuss. The published RNAseq data suggested to be relevant (Fig S1, refs 44-48) is not actually relevant to starvation/refeeding in the gut. To make their point and validate their reporter, the authors really need to at least present some qRT-PCR or in situ hybridization data that documents *Acbp6* mRNA expression in the gut during feeding/fasting transitions. Moreover, reporter gene and mRNA data don't necessarily reflect true levels of protein expression or activity. Ideally, the authors would measure *Acbp6* protein expression using an antibody (which they could make), and also assay its activity somehow, perhaps by measurements of bound Acyl-CoA. Without such additional data the paper grossly overstates its conclusions about the regulation of metabolism by *Acbp6*. As it stands, the data show that the gene is important for adaptive plasticity, but do not show that changes in *Acbp6* levels or activity occur, or that such changes regulate changes in metabolism needed for adaptive plasticity.

We recognize the importance of validating *Acbp6*-RFP reporter activity as an accurate readout of *Acbp6* mRNA expression. To this end, we devised a strategy to measure *Acbp6* transcript in the posterior midgut specifically, first dissecting out the entire midgut and then further dissecting the posterior midgut in order to extract RNA specifically from this region of the intestine. Using this dissection method, we were able to show that *Acbp6* transcription (via qRT-PCR) is induced during fasting and begins to normalize during refeeding (*see Supplemental Figure 2e and below*). These data mirror *Acbp6*-RFP reporter activity during fasting-refeeding transitions.

Generating antibodies against *Drosophila* proteins that are part of large gene families (and have similar sequences) is not trivial. However, since the last submission we finished generating an *Acbp6^P-ACBP6^{V5}* transgenic fly expressing a V5-tagged ACBP6 protein under the control of an *Acbp6* promoter/enhancer region (*Acbp6^P*, similar to the RFP reporter). Immunostaining confirmed that ACBP6-V5 protein level is up-regulated during fasting (with localization in the cytoplasm of posterior midgut enterocytes) and normalized upon refeeding (*see Supplemental Figure 2f, g and below*).

While we admit these data don't provide the same information as an endogenous antibody, but they do further support a model of dynamic/inducible regulation of *Acbp6* in the *Drosophila* midgut during nutrient adaptation.

2. Another point relates to the acetyl-lysine assays shown in Fig 4 and Fig S9C. These are the only link in the paper to Acyl-CoA metabolism apart from the *Acbp6* sequence homology. From their data the authors argue that levels of total protein

acetylation decrease upon fasting. However, the data presented are either pictures, or measures of numbers of “Acetylated Lysine +” cells, or “Acetylated Lysine area”. These metrics cannot be accurate reflections of the amount of protein acetylation per cell, or per unit chromatin, or per unit protein. Although the pictures look good, I would ask the authors to re-do their quantifications, measuring something that is a more accurate metric for how much total acetylation is present relative to a standard such as total protein, total DNA, or total cell volume or cell area. For instance, integrated GFP intensity/area would do. Given the way the samples were assayed, I didn’t find Fig 4b, c, e or S9c convincing. It should be straightforward for the authors to re-do their quantifications using a different approach, perhaps even using the same images. This could better support their proposal that fasting and *Acbp6* affect an output of Acyl-CoA metabolism.

As the reviewer recommended, we have re-done all quantifications of pan-lysine protein acetylation in Fig. 4 and Supplemental Figure 9 (9c is now 9e) using an integrated (Fluor.) intensity / area analysis of images. These quantifications show similar statistically significant differences in immunostaining (fluor. intensity) related to pan lysine acetylation in the posterior midgut (see Figure 4c, e and Supplemental Figure 9e and below).

3. I still suggest changing the Title. Neither this paper, nor any other, has actually shown that *Drosophila* *Acbp6* binds Acyl-CoA, yet the title asserts this. In fact, *Acbp6* is just ~50% conserved with its orthologs, ACBD7 and DBI, so there is a possibility it has some other molecular function.

We don’t necessarily disagree with the suggestion. However, this is tricky since Acyl-CoA binding is in the name of the gene (and justifiably so based on the high sequence/domain conservation across taxa (and even kingdoms) after hundreds of millions of years of evolution). The title is meant to reflect the gene family, and a specific gene, as opposed to function...but it’s hard to avoid that due to its gene name. To this end, we have added the actual gene name (*ACBP6*) in parenthesis after - ‘A distinct Acyl-CoA binding protein (*ACBP6*)’ - to further reflect that we are talking about genes/gene families and not necessarily function.

Reviewer #3 (Remarks to the Author):

The authors have been very responsive to the original critiques, and the manuscript is now much stronger. I have no additional remarks.

REVIEWERS' COMMENTS

Reviewer #2 (Remarks to the Author):

The authors have now addressed all of this reviewer's criticisms, by the inclusion of substantial and excellent new data, and some minor editorial changes. This is quite a good paper and is very appropriate for publication.